# NONO, SFPQ, and PSPC1 promote telomerase recruitment to the telomere

Alexander P. Sobinoff [1], Jadon K. Wells [1], Maurice Chow[1], Christopher B. Nelson[1], Xinyi Wu[2], Scott B. Cohen[3], Yu Heng Lau [2], Tracy M. Bryan [3], Archa Fox [4] & Hilda A. Pickett [1] ✉

Telomerase is a ribonucleoprotein enzyme that maintains telomeric repeats on chromosome ends in continuously dividing cells. Telomere maintenance via telomerase is dependent on the correct assembly of the enzyme complex, complex stabilization by associated cofactors, and effective recruitment to the telomere. Here, we show that telomerase is regulated in each of these processes by the Drosophila behaviour/human splicing (DBHS) family of RNA/DNA binding proteins (NONO, SFPQ and PSPC1). The DBHS proteins associate with catalytically active telomerase through the hTR RNA template component. Cells lacking the DBHS proteins display telomerase retention in nuclear Cajal bodies and impaired telomerase recruitment to the telomere, with NONO and PSPC1 depletion culminating in progressive telomere shortening in several cell lines, with the exception of long-term NONO depletion in 293 and 293T. Our results reveal the DBHS protein family as components of the telomerase trafficking machinery integral to telomere maintenance.

Human chromosomes are capped by telomeres, specialised DNA-protein structures that are crucial for genome stability[1]. Telomeres shorten during each cellular division, with cumulative attrition resulting in telomeric damage and replicative senescence[2]. Continuously dividing cells maintain their telomeres through telomerase, a ribonucleoprotein (RNP) enzyme complex consisting of the catalytic reverse transcriptase hTERT and the RNA template component hTR. Germline mutations in telomerase associated genes cause premature telomere shortening, a condition characterised by pulmonary fibrosis, aplastic anemia, bone marrow failure and liver cirrhosis[3]. Telomerase activation is also an essential part of oncogenesis, with recurrent promoter mutations that upregulate hTERT being the most frequent mutation in various cancer subtypes[4]. Despite the critical importance of telomerase regulation in several human pathologies, our understanding of telomerase biology remains incomplete.

Telomerase associates with several accessory proteins, which are essential to form the active enzyme complex. A number of these proteins interact with the complex via the RNA template component hTR. The RNA binding proteins dyskerin, NOP10, NHP2, and GAR1 form a complex with hTR to protect the RNA subunit from degradation[5]. The hTR component also contains a Cajal body box (CAB box), comprising the terminal hairpin loop of the scaRNA domain, that facilitates trafficking of telomerase to subnuclear membrane-less organelles called Cajal bodies by the WD40 repeat protein TCAB1 (WRAP53)[6,7]. Telomerase is released from Cajal bodies and recruited to the telomere during S-phase, engaging in many probing interactions with the telomere before forming a stable association with the shelterin component TPP1[8–10]. Following recruitment to the telomere, the single-stranded 3' G-rich overhang acts as a substrate for telomerase, annealing to the RNA template motif within hTR, and enabling hTERT to reverse transcribe successive telomeric repeats onto the chromosome terminus[11].

The DBHS family are highly conserved RNA/DNA binding proteins that localise to specialised subnuclear organelles, chromatin, and DNA damage foci[12]. In humans, there are three members of the family, NONO, SFPQ, and PSPC1. All three DBHS proteins share a highly conserved core of ~300 amino acids defined as the 'DBHS region', which

[1]Telomere Length Regulation Unit, Children's Medical Research Institute, Faculty of Medicine and Health, The University of Sydney, Westmead, NSW, Australia. [2]School of Chemistry, The University of Sydney, Camperdown, NSW, Australia. [3]Cell Biology Unit, Children's Medical Research Institute, Faculty of Medicine and Health, The University of Sydney, Westmead, NSW, Australia. [4]School of Human Sciences, The University of Western Australia, Perth, WA, Australia. ✉e-mail: hpickett@cmri.org.au

regulates RNA binding, protein/nucleic acid interactions, dimerization, and polymerization. DBHS proteins are fundamentally dimeric, forming homo- or heterodimers that can readily exchange their interaction partner to form an alternative complex to ensure a degree of functional redundancy[13]. Functionally, DBHS proteins are involved in several aspects of RNA processing and DNA repair[13–15]. NONO and SFPQ have also both been found to associate with the telomere during replication, and can interact with telomere repeat containing RNA, or TERRA, to prevent the formation of RNA:DNA hybrids at telomeres[16,17].

Here, we demonstrate that the DBHS proteins directly interact with telomerase via the hTR component to facilitate trafficking of telomerase out of Cajal bodies and its recruitment to the telomere. Our data suggest distinct roles for individual DBHS members in telomerase regulation, with each protein being separately critical for telomerase recruitment. We specifically identify that the DBHS region regulates telomerase recruitment to the telomere and trafficking out of Cajal bodies, and that SFPQ and PSPC1 polymerization is essential for cell viability. Finally, we show that long term NONO and PSPC1 depletion results in progressive telomere shortening in cancer cells.

## Results

### DBHS proteins interact with active telomerase via hTR

To identify RNA binding proteins positively associated with telomere length, we used the publicly available Cancer Dependency Map Portal (DepMap Public 23Q4) to determine the linear regression between gene expression (Expression Public 23Q3) and telomere length (TelSeq Telomere Content; CCLE_WGS_Telseq_log2) for 402 RNA binding proteins (RBPDB; http://rbpdb.ccbr.utoronto.ca/), the Cajal body component *COIL*, and the shelterin complex (*TERF2, TERF1, RAP1, ACD, TIN2* and *POT1*) in 325 cancer cell lines (Supplementary Fig. 1A). We found that expression of the catalytic component of telomerase, *hTERT*, was most significantly correlated with telomere length, validating this approach. The second highest ranked RNA binding protein was *NONO*, followed closely by telomerase associated *HNRNPA1* and *HNRNPH3*[18]. Other positive hits previously associated with telomere length were *HNRNPD, PCBP2* and *TOE1*[19–22]. The other DBHS proteins, SFPQ and PSPC1, were also positively associated with telomere length, although they did not meet the significance threshold. These data suggest a positive role for the DBHS proteins in regulating telomere length, potentially through telomerase.

To explore the potential role for NONO and the other DBHS proteins in regulating telomerase biology, we selected several telomerase positive cell lines (293T, 293, HT1080, HCT116, HeLa) with relatively consistent DBHS expression and variable telomere lengths for investigation (Supplementary Fig. 1B, C). Immunoprecipitation (IP) experiments revealed all three DBHS proteins associate with hTR (Fig. 1A). Electrophoretic mobility shift assays (EMSAs) using immunopurified exogenously expressed DBHS proteins demonstrated concentration-dependent binding of recombinant NONO, SFPQ, and PSPC1 to α-32P-GTP radiolabelled hTR probe indicated by a gel shift pattern (Fig. 1B; Supplementary Fig. 1D). A super-shift was also observed in the presence of antibodies against immunopurified DBHS proteins, supporting the specificity of their binding to hTR. The DBHS proteins were also present in hTERT IP experiments, suggesting that they may interact with the active telomerase RNP complex (Fig. 1C). To test this, we stably overexpressed NONO-Myc/FLAG in 293 and HT1080 cells and performed a Myc IP followed by a Telomere Repeat Amplification Protocol (TRAP) assay to measure telomerase activity[23]. A positive TRAP signal was detected in the eluate of both 293 and HT1080 cells, suggesting that the DBHS proteins interact with the active telomerase RNP complex (Fig. 1D). A positive TRAP signal was also detected in the eluate of 293T cells stably overexpressing SFPQ-Myc/FLAG and PSPC1-Myc/FLAG (Supplementary Fig. 1E).

The DBHS proteins mediate a wide variety of protein-protein and protein-nucleic acid interactions, acting as a multipurpose molecular scaffold[12]. To determine how the DBHS proteins interact with the telomerase RNP complex, we overexpressed wild-type (Wt) hTR and hTERT separately in ALT positive U-2 OS cells, which do not endogenously express these components of telomerase. As in telomerase positive cells, all three DBHS proteins interact with Wt hTR (Fig. 1E). NONO has been shown to interact with RNA G-quadruplex motifs like those observed in the first 20 nucleic acids of hTR[24,25]. However, deletion of this region did not affect NONO/hTR binding (Supplementary Fig. 1F). Interestingly, the DBHS proteins only interacted with overexpressed hTERT in the presence of hTR in U-2 OS cells (Fig. 1F). This suggests that the DBHS proteins interact with the active telomerase RNP complex via hTR.

### DBHS proteins promote telomerase trafficking from Cajal bodies to the telomere

Telomerase localizes within Cajal bodies and is recruited to the telomere maximally in mid S-phase[5]. As the DBHS proteins are involved in RNA trafficking and act as a molecular scaffold for protein-nucleic acid interactions, we performed fluorescence in situ hybridization (FISH) for hTR and telomeres together with immunostaining for coilin, a key component of Cajal bodies, in S-phase DBHS protein depleted cells (Fig. 1G). Analysis of telomerase trafficking revealed significantly less telomerase recruitment to the telomere in DBHS protein depleted 293T and HT1080 cells compared to scrambled control (Fig. 1H; Supplementary Fig. 1G). Analysis of telomerase trafficking in cells treated with a control siRNA (scrambled) revealed that hTR was not present in Cajal bodies in most mid S-phase cells (Fig. 1I; Supplementary Fig. 1H). In contrast, loss of both NONO and SFPQ caused a substantial increase in hTR positive Cajal bodies. No change in hTR positive Cajal bodies was observed after PSPC1 depletion. Taken together, these data show that DBHS protein loss impairs telomerase recruitment to telomeres, and that NONO/SFPQ depletion causes telomerase retention in Cajal bodies.

Unlike NONO and SFPQ, PSPC1 localisation has not previously been reported at the telomere[16]. Telomere FISH coupled with immunostaining revealed all three DBHS protein members associate with telomeres in 293T and HT1080 cells (Supplementary Fig. 2A, B). The DBHS proteins were also found to associate with coilin foci, albeit at a lower frequency (Supplementary Fig. 2C).

In addition to their multitude of roles in RNA processing and DNA repair, the DBHS proteins also associate with the long non-coding RNA NEAT1 to form paraspeckles[12]. To determine whether paraspeckles are involved in telomerase biology, we observed NEAT1/hTR co-localisations via FISH. NEAT1 rarely associated with hTR in 293T cells (Supplementary Fig. 2D). However, NEAT1 did associate with telomeres in multiple cell lines, suggesting a potential role for paraspeckles at the telomere (Supplementary Fig. 2E).

The DBHS protein family has several roles in regulating gene expression via transcription, splicing, and RNA transport[12]. Western blot analysis revealed that short term NONO and SFPQ loss does not impact the expression of proteins associated with telomerase recruitment/function (NHP2, GAR1, Dyskerin, NOP10, reptin, pontin, TCAB1, coilin) (Supplementary Fig. 1I). However, short term PSPC1 depletion did result in reduced coilin protein, mRNA transcript, and foci (Supplementary Figs. 1J, K). PSPC1 IP revealed no direct interaction between PSPC1 and coilin RNA, suggesting PSPC1 is not directly involved in coilin mRNA processing (Supplementary Fig. 1L). The loss of coilin accounts for the lack of hTR positive Cajal bodies observed after PSPC1 depletion (Fig. 1I) and suggests different roles for individual DBHS proteins in regulating telomerase recruitment. Depletion of individual DBHS proteins did not affect TPP1 or TIN2 protein expression (Supplementary Fig. 1M).

TCAB1 is an RNA chaperone that is required for the recruitment of hTR to Cajal bodies[6,7]. As NONO and SFPQ depletion causes retention of hTR in Cajal bodies, indicative of a role for NONO and SFPQ in

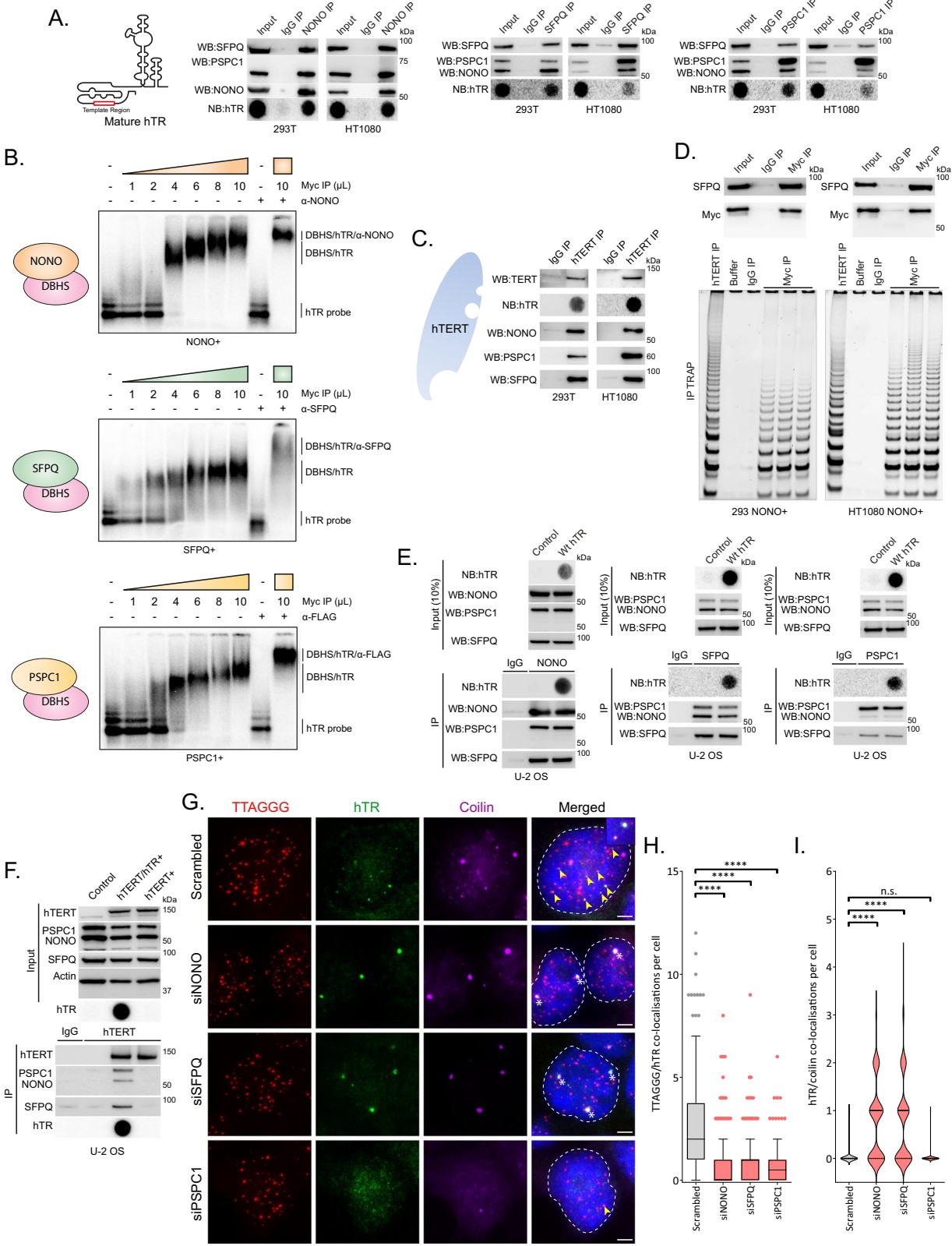

trafficking hTR out of Cajal bodies, we examined whether the DBHS proteins compete with TCAB1 for hTR binding. TCAB1 depletion did not influence the ability of NONO to bind hTR (Supplementary Fig. 3A). In contrast, PSPC1 protein depletion resulted in decreased total coilin/TCAB1 co-localisations (Supplementary Fig. 3B), but this difference was not seen when normalised to total coilin foci (Supplementary Fig. 3C), consistent with reduced coilin expression (Supplementary Fig. 1I). Depletion of coilin did not impact NONO binding to hTR,

suggesting NONO does not require Cajal bodies to interact with hTR. (Supplementary Fig. 3D).

## Cell line specific effects of DBHS protein depletion on hTR accumulation and telomerase assembly

To determine whether DBHS protein loss also impacts telomerase assembly, we immunopurified endogenous telomerase hTERT and examined the levels of total and hTERT associated hTR

**Fig. 1 | DBHS proteins interact with telomerase RNP via hTR and promote telomerase recruitment to telomeres. A** Immunoprecipitation (IP) of DBHS proteins with hTR in 293T and HT1080 cells ($n = 3$). Indicated proteins assayed by Western blot; hTR assayed by northern dot blot. **B** EMSA showing DBHS proteins binding to radiolabelled hTR. Overexpressed DBHS proteins were immunopurified using an anti-Myc antibody ($n = 2$). Bound complex is indicated by gel shift. Supershift complex comprised of hTR, DBHS proteins and α-DBHS or α-FLAG antibody is indicated by a further shift. **C** Interaction of DBHS proteins with immunopurified hTERT in 293T and HT1080 cells ($n = 3$). Indicated proteins assayed by Western blot; hTR assayed by northern dot blot. **D** TRAP on immunopurified overexpressed NONO (anti-Myc antibody; $n = 3$). Indicated proteins assayed by Western blot; immunopurified hTERT used as a positive control. Buffer only (buffer) and IgG IP were used as negative controls. **E** DBHS protein interaction with overexpressed hTR in hTERT negative U-2 OS cells transfected with plasmids expressing wild-type (Wt) hTR and DKC1 ($n = 3$). Input, indicated proteins assayed by Western blot; hTR assayed by northern dot blot; IP, DBHS proteins and hTR immunopurified using

NONO, SFPQ, and PSPC1 antibodies assayed by Western or northern dot blot. IgG IP, non-specific IgG used for IP. **F** DBHS protein interaction with overexpressed hTERT in hTR negative U-2 OS cells stably transfected with plasmids expressing hTERT and/or Wt hTR and DKC1 ($n = 3$). Input, indicated proteins assayed by Western blot. IP, DBHS proteins and hTERT immunopurified using hTERT antibodies assayed by Western blot. IgG IP, non-specific IgG used for IP.
**G** Representative images of telomere (red), hTR (green), and coilin (purple) co-localizations in DBHS depleted 293T cells. TTAGGG/hTR co-localizations are indicated by yellow arrows; hTR/coilin co-localizations are indicated by white asterisks. Scale bars are 5 µm. **H** Tukey boxplots (median, interquartile range, and Tukey whiskers) of TTAGGG/hTR co-localizations in DBHS depleted 293T cells. Out of three experiments, $n = 204$ cells scored per treatment, ****$p < 0.0001$, Kruskal-Wallis test. **I** Violin plot of hTR/coilin co-localizations in DBHS depleted 293T cells. Out of three experiments, $n = 150$ cells scored per treatment, n.s. = non-significant, ****$p < 0.0001$, Kruskal-Wallis test. Source data are provided as a Source data file.

(Supplementary Fig. 4A). Depletion of either NONO or SFPQ caused a significant decrease in total hTR levels (input) in HT1080 cells, but did not affect hTR levels in 293T cells (Supplementary Fig. 4B). DBHS protein depletion did not significantly influence the amount of immunopurified hTERT in HT1080 and 293T cells, suggesting the DBHS proteins do not influence hTERT expression (Supplementary Fig. 4C). However, both NONO and SFPQ depletion noticeably increased the amount of co-immunoprecipitated hTR in the 293T hTERT eluate (Supplementary Fig. 4D). As DBHS protein loss did not affect global hTR or hTERT levels in 293T cells, this suggests that NONO and SFPQ depletion leads to more efficient assembly of the catalytic core of telomerase in this cell line. This was confirmed by an increase in telomerase activity measured by qTRAP (Supplementary Fig. 4E). Conversely, DBHS protein depletion caused a significant decrease in the amount of assembled telomerase RNP in HT1080 cells (Supplementary Fig. 4D). This was confirmed by qTRAP, and trends closely with the amount of total hTR (input) in HT1080 DBHS depleted cells (Supplementary Fig. 4E;4B). Overall, these data show that DBHS protein loss has opposing effects on the accumulation of hTR and telomerase RNP assembly in 293T and HT1080 cells.

## NONO/PSPC1 overexpression rescues telomerase trafficking in SFPQ depleted cells
The DBHS proteins are characterised by the highly conserved DBHS region. Given the homology between the three proteins, we hypothesised that overexpression of one family member could compensate for the loss of another. We examined whether overexpressing one DBHS protein could compensate for the loss of another in telomerase recruitment assays in both 293T and HT1080 cells (Fig. 2A; Supplementary Fig. 5A, B). Overexpression of individual DBHS proteins was unable to rescue the telomerase recruitment deficit caused by the loss of another (Fig. 2B–D; Supplementary Fig. 5C–E). However, NONO overexpression did partially rescue telomerase retention within Cajal bodies caused by SFPQ depletion (Fig. 2B; Supplementary Fig. 5C). PSPC1 overexpression was also able to restore telomerase trafficking out of Cajal bodies in SFPQ depleted cells (Fig. 2C; Supplementary Fig. 5D). However, SFPQ overexpression could not compensate for deficits in telomerase trafficking caused by NONO and PSPC1 depletion (Fig. 2D; Supplementary Fig. 5E). Overexpression of NONO and SFPQ was unable to rescue coilin expression in PSPC1 depleted cells (Supplementary Fig. 5B). These results suggest that the defect in telomerase recruitment caused by DBHS protein depletion is not solely due to telomerase retention within Cajal bodies, that each of the DBHS proteins are separately critical for telomerase recruitment, and that the role of SFPQ in telomerase trafficking out of Cajal bodies can be substituted by NONO and PSPC1. The observation that PSPC1 can rescue telomerase trafficking

in SFPQ depleted cells also suggests a role for PSPC1 in telomerase trafficking beyond regulating coilin expression.

The ability of NONO and PSPC1 to rescue trafficking of telomerase out of Cajal bodies, but not telomerase recruitment to the telomere, suggests that all three DBHS members may be required to interact with the telomere to facilitate telomerase recruitment. To investigate this, we performed chromatin immunoprecipitation (ChIP) to observe DBHS protein association with the telomere when individual members of the family were depleted (Fig. 2E; Supplementary Fig. 5F). In scrambled control cells, all three DBHS proteins associate with the telomere, with NONO and SFPQ being the strongest interactors. Depletion of both SFPQ and PSPC1 significantly reduced the association of other DBHS proteins with the telomere, while NONO depletion reduced SFPQ association (Fig. 2E). Unlike telomere association, depleting one member of the DBHS family did not impair the ability of the others to interact with telomerase, as measured by hTERT immunoprecipitation (Fig. 2F). This suggests that the presence of all three DBHS proteins is required for their optimal telomere association, but not for their association with telomerase, potentially explaining why overexpression of NONO and PSPC1 can compensate for the loss of SFPQ in telomerase trafficking out of Cajal bodies, but not in telomerase recruitment to the telomere.

## A functional DBHS region is crucial for telomerase recruitment
The DBHS region encompasses two unique RNA recognition motifs (RRM1, RRM2), the protein–protein interaction NonA/paraspeckle domain (NOPS), and the coiled-coil domain (Fig. 3A). DBHS RRM1 is a canonical RRM domain consisting of four anti-parallel β-strands and two α-helices arranged in a β1α1β2β3α2β4 fold that interacts with single-stranded RNA[12,26]. However, the structure of the DBHS RRM2 domain is unique, closely resembling a double-stranded DNA/RNA recognition motif, and providing a scaffold for DBHS protein dimerization[12,26]. The NOPS domain is primarily involved in DBHS protein dimerization, and the highly charged coiled-coil domain facilitates dimerization and oligomerization[27].

To determine which functional domains of the DBHS region are important for telomerase recruitment, we engineered a panel of four mutant constructs in which each domain of the DBHS region was either deleted (ΔRRM1 and ΔRRM2) or disrupted (ΔPol and ΔDim) (Fig. 3A). The ΔPol mutant has a quadruple alanine substitution within the coiled-coil domain which has been previously shown to disrupt the self-association of DBHS dimers into higher-order structures through oligomerization[28]. The ΔDim mutant has a double alanine substitution within the NOPS domain that is critical for DBHS dimerization[26]. These constructs were stably transduced into 293T and HT1080 cells, with exogenous expression of NONO (n.ΔRRM1, n.ΔRRM2, n.ΔPol, n.ΔDim), SFPQ (s.ΔRRM1, s.ΔRRM2, s.ΔPol, s.ΔDim), and PSPC1 (p.ΔRRM1,

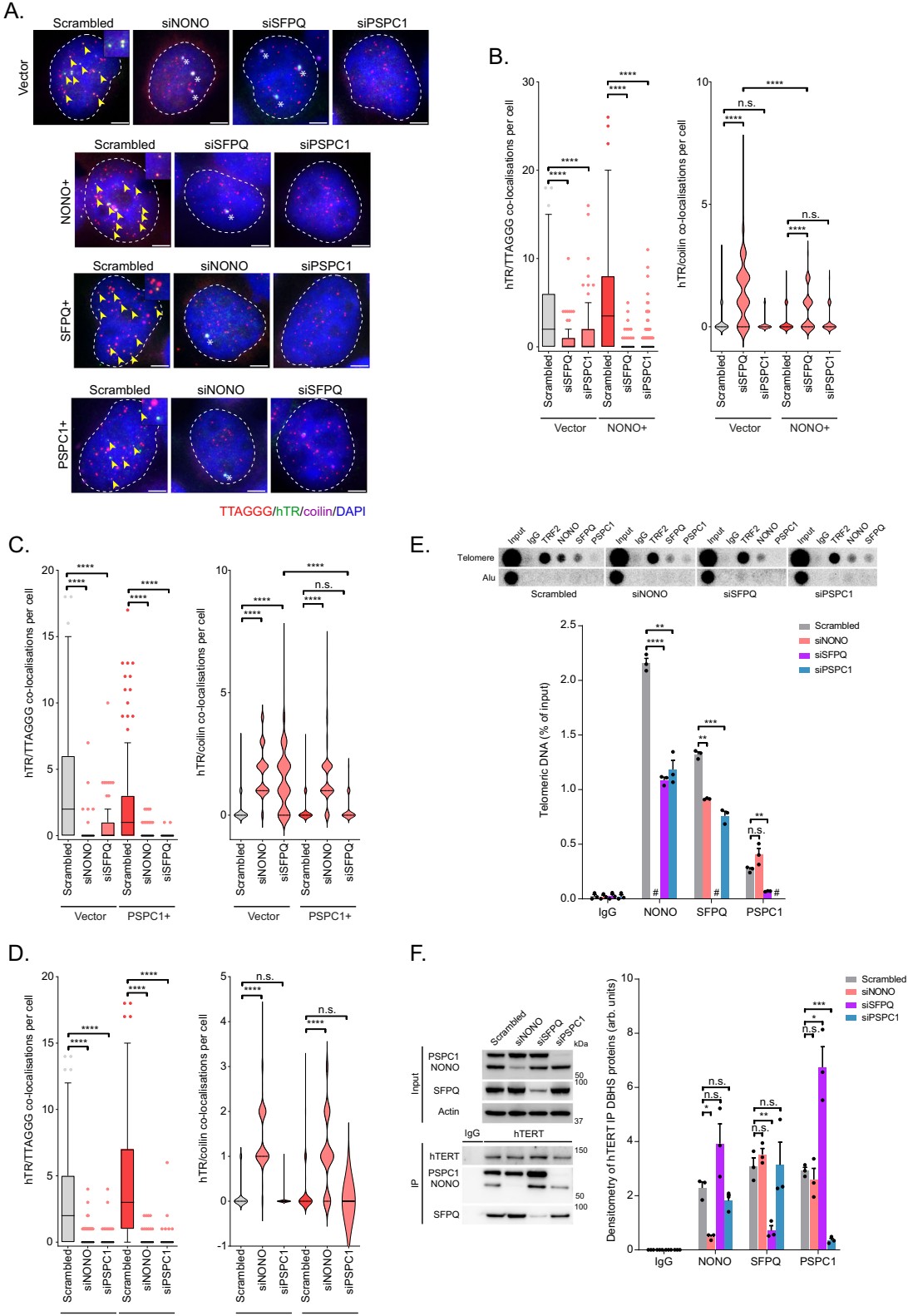

p.ΔRRM2, p.ΔPol, p.ΔDim) mutants confirmed by Western blot analysis (Supplementary Fig. 6A). We then performed rescue experiments on NONO and SFPQ wild-type (NONO+; SFPQ+) and functional mutant expressing cells through siRNA depletion (Supplementary Fig. 6B). For PSPC1, rescue experiments were performed by stably overexpressing wild-type (PSPC1+) and functional mutants in PSPC1 CRISPR knockout (KO) cells (Supplementary Fig. 6C).

Overexpression of wild-type DBHS proteins rescued the effects of endogenous protein depletion on telomerase recruitment and trafficking out of Cajal bodies in 293T and HT1080 cells (Fig. 3B–E; Supplementary Fig. 6D–G). Most domain mutants for each DBHS protein failed to rescue telomerase recruitment and trafficking out of Cajal bodies in 293T and HT1080 cells (Fig. 3B–E; Supplementary Figs. 6D–G). Overexpression of n.ΔPol, p.ΔRRM1 and p.ΔDim did

**Fig. 2 | NONO/PSPC1 overexpression partially rescues siSFPQ induced hTR sequestration in Cajal bodies. A** Representative images of telomere (red), hTR (green), and coilin (purple) co-localizations in DBHS depleted 293T cells over-expressing NONO (NONO+), SFPQ (SFPQ+) or PSPC1 (PSPC1+). TTAGGG/hTR co-localizations are indicated by yellow arrows; hTR/coilin co-localizations are indicated by white asterisks. Scale bars are 5 μm. **B–D** Tukey boxplots (median, inter-quartile range, and Tukey whiskers) of TTAGGG/hTR co-localizations in NONO+, SFPQ+, or PSPC1 + DBHS protein depleted 293T cells (left panels). Out of three experiments, $n = 150$ cells scored per treatment, ****$p < 0.0001$, Kruskal-Wallis test. Violin plots of hTR/coilin co-localizations in NONO+, SFPQ+, or PSPC1 + DBHS protein depleted 293T cells (right panels). Out of three experiments, $n = 150$ cells scored per treatment, n.s. = non-significant, ****$p < 0.0001$, Kruskal-Wallis test.

**E** Telomere-ChIP against DBHS proteins in 293T cells after DBHS protein knock-down. Values are mean ± SEM from $n = 3$ experiments, n.s. = non-significant, **$p = 0.0019$ between NONO Scrambled and siPSPC1, $p = 0.0023$ between SFPQ Scrambled and siNONO, $p = 0.0045$ between PSPC1 Scrambled and siSFPQ, ***$p = 0.0008$, ****$p < 0.0001$, # = no data, Welch's two-tailed $t$-test. **F** Interaction of DBHS proteins with hTERT assayed by IP of hTERT in 293T cells after DBHS protein knockdown (left panel). Indicated proteins assayed by Western blot. Quantitation of DBHS proteins in hTERT IP samples (right panel). Values are mean ± SEM from $n = 3$ experiments, n.s. = non-significant, *$p = 0.0103$ between NONO Scrambled and siNONO, $p = 0.0344$ between SFPQ Scrambled and siSFPQ, **$p = 0.0059$, ***$p = 0.0003$, Welch's two-tailed $t$-test. Source data are provided as a Source data file.

restore telomerase recruitment in HT1080 cells (Supplementary Fig. 6D; Supplementary Fig. 6G). Overexpression of s.ΔRRM2 was also able to rescue telomerase trafficking out of Cajal bodies in 293T cells (Fig. 3C). Interestingly, only overexpression of p.ΔRRM2 restored tel-omerase trafficking out of Cajal bodies in both 293T and HT1080 gPSPC1 cells (Fig. 3D; Supplementary Fig. 6G).

Several of the DBHS protein domain mutants also exhibited a dominant-negative effect on recruitment in Scrambled control cells, including n.ΔRRM2, s.ΔRRM1 and s.ΔPol in 293T cells (Fig. 3B, C). Stable overexpression of PSPC1 mutants p.ΔRRM1, p.ΔRRM2, and p.ΔDim all caused a reduction in telomerase recruitment in 293T, while all four functional mutants impaired telomerase trafficking out of Cajal bodies, suggesting a dominant-negative effect (Fig. 3D). Wild type PSPC1 and p.ΔRRM1 also had a dominant-negative effect on telomer-ase recruitment in HT1080 cells (Supplementary Fig. 6F). Interestingly, knockdown of endogenous SFPQ in HT1080 cells stably over-expressing s.ΔRRM1 and s.ΔPol resulted in lethality (Supplementary Fig. 6E). Overexpression of p.ΔPol was also lethal in both 293T and HT1080 gPSPC1 CRISPR cells (Fig. 3E; Supplementary Fig. 6G).

Overall, we demonstrate all four domains of the DBHS region are essential for DBHS protein mediated telomerase recruitment. However, only the NONO and SFPQ DBHS regions appear to be indi-vidually essential for telomerase trafficking, as the PSPC1 p.ΔRRM2 functional mutant reduced telomerase retention in Cajal bodies in gPSPC1 293T and HT1080 cells. The data also indicate dominant-negative effects on telomerase recruitment to the telomere following overexpression of either s.ΔRRM1 or s.ΔPol in 293T Scrambled control cells, and lethality following overexpression of these mutants in HT1080 siSFPQ cells. Given p.ΔPol overexpression in gPSPC1 CRISPR knockout cells was also lethal, these results suggest an essential role for SFPQ and PSPC1 mediated polymerisation of DBHS proteins for cell survival.

### Chemical inhibition of NONO impairs telomerase recruitment in cells

Controlling telomerase recruitment to the telomere has important implications for the treatment of several human pathologies, including cancer. (R)-SKBG-1 is an electrophilic small molecule that covalently binds NONO C145, resulting in the formation of dysfunctional nuclear aggregates and impaired processing of mRNA in cancer cells[29]. This amino acid is unique to NONO and is located within the hinge region between the RMM1 and RMM2 DBHS domains. To test whether telo-merase trafficking is affected by chemical inhibition of NONO, we treated 293T cells with (R)-SKBG-1 and its enantiomer (S)-SKBG-1 for 24 h and assayed telomerase recruitment (Supplementary Fig. 7A). (R)-SKBG-1 and (S)-SKBG-1 both significantly reduced levels of telomerase recruitment compared to the DMSO control group in 293T cells, while a significant reduction was only observed in (R)-SKBG-1-treated HT1080 cells (Supplementary Fig. 7B). Recruitment was significantly lower in (R)-SKBG-1-treated cells compared to (S)-SKBG-1, and the medians were the same between DMSO and (S)-SKBG-1 in 293T and HT1080-treated cells. Telomerase trafficking was not significantly

affected by either (R)-SKBG-1 or (S)-SKBG-1, similar to results seen in n.ΔRRM2 293T and n.ΔRRM1 HT1080 rescue experiments (Supple-mentary Fig. 7C; Fig. 3B; Supplementary Fig. 6D). (R)-SKBG-1 binds close to these two regions, potentially explaining why (R)-SKBG-1 treatment phenocopies these mutants. Treatment with (R)-SKBG-1 caused a slight but significant decrease in coilin foci in HT1080, but not 293T cells (Supplementary Fig. 7D). Coilin integrated intensity was also significantly reduced by (R)-SKBG-1 and (S)-SKBG-1 in 293T and HT1080 cells (Supplementary Fig. 7E). Treatment with (R)-SKBG-1 also caused a cell line specific effect on hTR levels similar to those caused by siNONO depletion (Supplementary Fig. 7E, Supplementary Fig. 4B). These results serve as a proof of concept that NONO can be chemically inhibited to reduce telomerase recruitment.

### NONO and PSPC1 depletion causes cancer cell specific changes in telomere length

The reduced telomerase recruitment and telomerase trafficking out of Cajal bodies observed with DBHS knockdown, coupled with the cell line specific fluctuations in telomerase activity, suggest that the DBHS proteins are key regulators of telomerase biology. A previous study has indicated that transient siRNA depletion of SFPQ reduces telomere foci intensity in the telomerase positive cell line H1299[16]. To understand the long-term effects of DBHS protein depletion on telomere length and telomerase biology, we generated several NONO and PSPC1 CRISPR KO cell lines. We were unable to generate SFPQ CRISPR KO cells due to associated lethality. HT1080 NONO knockout cell lines displayed progressive telomere shortening over multiple population doublings (PDs) (Fig. 4A). This was accompanied by elevated levels of DKC1 and NOP10 expression, two genes essential for hTR stability, which were not sufficient to maintain telomere length. Knockout of PSPC1 in HT1080 cells also caused progressive telomere shortening (Fig. 4B). Interestingly, long term PSPC1 depletion in HT1080 cells resulted in elevated levels of the telomerase trafficking protein TCAB1 and Cajal body component coilin. Knockout of PSPC1 in 293T cells also resulted in progressive telomere shortening and elevated coilin levels (Sup-plementary Fig. 8A). These results are in stark contrast to transient depletion, which resulted in reduced coilin expression in both cell lines. This suggests that either these cells have adapted to PSPC1 loss by elevating expression during selection, or that cells with abnormally high levels of TCAB1 and coilin provided a selection advantage when generating these lines. Given the loss in telomere length observed in these cells, these data also suggest that the effect of PSPC1 on telomere length is independent of Cajal bodies.

Analysis of telomerase recruitment revealed that both long term NONO and PSPC1 depletion impaired telomerase recruitment to the telomere in HT1080 cells, correlating with their shortened telomere length (Fig. 4C). Interestingly, some but not all NONO and PSPC1 knockout clones exhibited impaired telomerase trafficking out of Cajal bodies. Telomerase recruitment and trafficking out of Cajal bodies was also reduced in 293T cells after long term PSPC1 depletion (Supple-mentary Fig. 8B). These data suggest that retention of telomerase within Cajal bodies is not the only cause of reduced telomerase

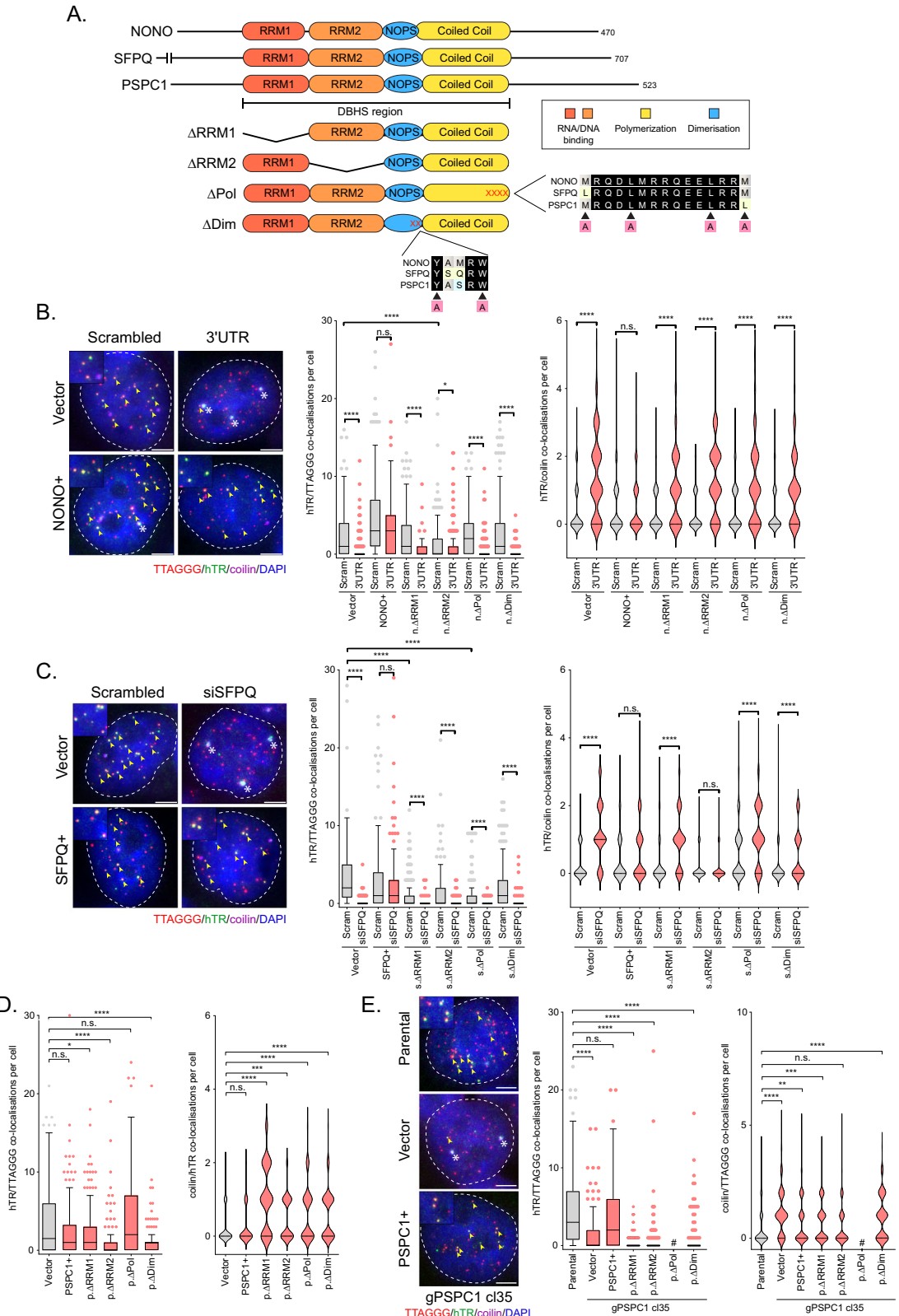

recruitment in DBHS depleted cells, and that HT1080 cells can partially adapt to long term depletion.

Surprisingly, 293T NONO knockout clones experienced a progressive increase in telomere length over time, with no change in the expression of telomerase associated proteins (Supplementary Fig. 8C). To determine whether this effect was cell line specific, we performed shRNA knockdown on several telomerase positive cell lines and

observed telomere length over successive PDs (Supplementary Fig. 8D). Stable shNONO knockdown in 293 and 293T cell lines resulted in increased telomere length, suggesting that this effect was not due to SV40 T-antigen transfection. In contrast, stable shNONO knockdown caused progressive shortening in HT1080, HeLa, and HCT116 cells. Reduced telomerase recruitment and hTR retention in Cajal bodies was also observed in stable shNONO knockdown HeLa and HCT116

**Fig. 3 | NONO, SFPQ, and PSPC1 mediated telomerase recruitment is attenuated by mutation of the DBHS region. A** Schematic of DBHS domain functions, mutations, or deletions. **B** Representative images of telomere (red), hTR (green), and coilin (purple) co-localizations in 293T 3'UTR NONO functional mutant rescue experiments (left panel). TTAGGG/hTR co-localizations are indicated by yellow arrows; hTR/coilin co-localizations are indicated by white asterisks. Scale bars are 5 μm. Tukey boxplots (median, interquartile range, and Tukey whiskers) of TTAGGG/hTR co-localizations (middle panel). Out of three experiments, $n$ = 198 cells scored per treatment, n.s. = non-significant, *$p$ = 0.0239, ****$p$ < 0.0001, Kruskal-Wallis test. Violin plots of hTR/coilin co-localizations (right panel). Out of three experiments, $n$ = 198 cells scored per treatment, n.s. = non-significant, ****$p$ < 0.0001, Kruskal-Wallis test. **C** Representative images of telomere (red), hTR (green), and coilin (purple) co-localizations in 293T siSFPQ functional mutant rescue experiments (left panel). TTAGGG/hTR co-localizations are indicated by yellow arrows; hTR/coilin co-localizations are indicated by white asterisks. Scale bars are 5 μm. Tukey boxplots (median, interquartile range, and Tukey whiskers) of TTAGGG/hTR co-localizations (middle panel). Out of three experiments, $n$ = 150 cells scored per treatment, n.s. = non-significant, ****$p$ < 0.0001, Kruskal-Wallis test. Violin plots of hTR/coilin co-localizations (right panel). Out of three

experiments, $n$ = 150 cells scored per treatment, n.s. = non-significant, ****$p$ < 0.0001, Kruskal-Wallis test. **D** Tukey boxplots (median, interquartile range, and Tukey whiskers) of TTAGGG/hTR co-localizations in 293T cells overexpressing PSPC1 functional mutants (left panel). Out of three experiments, $n$ = 150 cells scored per treatment, n.s. = non-significant, *$p$ = 0.0124, ****$p$ < 0.0001, Kruskal-Wallis test. Violin plots of hTR/coilin co-localizations in 293T cells overexpressing PSPC1 functional mutants (right panel). Out of three experiments, $n$ = 150 cells scored per treatment, n.s. = non-significant, ***$p$ = 0.0004, ****$p$ < 0.0001, Kruskal-Wallis test. **E** Representative images of telomere (red), hTR (green), and coilin (purple) co-localizations in 293T gPSPC1 cells overexpressing PSPC1 functional mutants (left panel). TTAGGG/hTR co-localizations are indicated by yellow arrows; hTR/coilin co-localizations are indicated by white asterisks. Scale bars are 5 μm. Tukey boxplots (median, interquartile range, and Tukey whiskers) of TTAGGG/hTR co-localizations (middle panel). Out of three experiments, $n$ = 150 cells scored per treatment, n.s. = non-significant, ****$p$ < 0.0001, Kruskal-Wallis test. Violin plots of hTR/coilin co-localizations (right panel). Out of three experiments, $n$ = 150 cells scored per treatment, n.s. = non-significant, **$p$ = 0.0033, ***$p$ = 0.0005, ****$p$ < 0.0001, Kruskal-Wallis test. Source data are provided as a Source data file.

cells, similar to that observed after siNONO depletion (Supplementary Fig. 8E; Fig. 1H, I). Stable shNONO depletion also caused a moderate decrease in hTR levels, which may contribute slightly to reduced telomerase recruitment (Supplementary Fig. 8F). NONO protein expression was not significantly different in 293 and 293T cells compared to HT1080, HCT116, and HeLa cells (Supplementary Fig. 1B). Despite having a significantly shorter parental telomere length compared to HT1080 and HeLa, 293 and 293T cell line telomere length was only slightly shorter than HCT116 cells, suggesting telomere length was not predictive of outcome in shNONO depleted cells (Supplementary Fig. 1C). This suggests that the change in telomere length following long term NONO depletion is not dependent on starting telomere length or DBHS protein expression and is cell line specific.

Although trafficking was reduced in NONO knockout 293T cells, telomerase recruitment was not significantly affected, explaining the lack of telomere shortening observed in these cells (Supplementary Fig. 8B). Telomerase activity was also not consistently affected by long term NONO depletion in 293T cells (Supplementary Fig. 9A), suggesting that changes in telomerase biology were not contributing to the observed increase in telomere length. Long term depletion of NONO resulted in a consistent decrease in telomerase activity in HT1080 cells comparable to short term siRNA depletion (Supplementary Fig. 9A; Supplementary Fig. 4E). Overall, the levels of hTR in the NONO and PSPC1 knockout cells do not positively correlate with the reduced telomerase recruitment, suggesting that telomere shortening is not solely due to lower hTR levels (Supplementary Fig. 9B). It should be noted that telomerase activity varies considerably between clonal populations in cancer cell lines[30]. Therefore, changes in telomerase activity observed between parental and DBHS knockout clones may be due to clonal variation rather than the loss of DBHS expression.

A previous report has suggested that NONO controls telomere length by supressing the formation of telomere RNA:DNA hybrids, with depletion of NONO causing increased TERRA levels, telomere RNA:DNA hybrids, and telomere replication stress[16]. Northern dot blotting revealed no changes in TERRA levels between parental controls and NONO knockout 293T cells (Supplementary Fig. 9C). Long term deletion of NONO in 293T cells also had no effect on the formation of replication stress-induced pRPA2(S33) colocalizations at telomeres, with NONO cl5 showing a decrease in overall pRPA2(S33) levels (Supplementary Fig. 9D). NONO depleted 293T cells were also negative for the Alternative Lengthening of Telomeres (ALT) biomarkers C-circles and ALT associated PML bodies (APBs) (Supplementary Fig. 9E, F), indicative of telomere lengthening not being the result of ALT activation.

Overall, our results suggest that long term PSPC1 depletion causes impaired telomerase recruitment, resulting in significant telomere shortening. Long term NONO depletion also impacts telomerase recruitment and telomere length, although the effects appear to be cell line dependent.

## Discussion

Telomerase and its telomere substrate are in exceptionally low abundance in cells, with a peak of ∼250 telomerase molecules and 184 telomeres in an immortal human cell in late S phase[31]. The tightly controlled recruitment of low-abundance telomerase to telomeres is therefore essential for telomere maintenance. Despite the importance of this process, our understanding of telomerase recruitment regulation is limited. Here, we provide the first evidence of the DBHS protein family as regulators of telomerase recruitment in cancer cells. NONO, SFPQ, and PSPC1 all interact with the active telomerase complex via the hTR RNA scaffold to promote efficient telomerase recruitment to the telomere. The three DBHS proteins form heterodimers and share over 70% identity within the DBHS region, introducing functional overlap and redundancy as evidenced by functional compensation for NONO loss by PSPC1 in DNA repair[13]. However, our data indicate distinct roles for the DBHS proteins in facilitating telomerase recruitment, evident by depletion of each individual DBHS protein impairing telomerase recruitment to the telomeres, and the lack of functional redundancy between the DBHS members. This suggests that the DBHS proteins play a coordinated role in telomerase regulation, with each protein fulfilling a distinct yet integrated function in the pathway.

Telomerase localises to Cajal bodies throughout most of the cell cycle and is released to facilitate recruitment to the telomeres during S-phase[8,32–34]. The physiological relevance of this process is contentious, but it is generally accepted that Cajal bodies make a major contribution to telomerase trafficking and recruitment to telomeres, with Cajal body disruption impairing telomerase recruitment[6,9,35–37]. Our data suggest both NONO and SFPQ facilitate telomerase trafficking out of Cajal bodies during S-phase, promoting telomerase recruitment. We have shown that chemical inhibition of NONO with (*R*)-SKBG-1 also impairs telomerase recruitment[29]. PSPC1 appears to play an additional role in telomerase trafficking by promoting the expression of the Cajal body component coilin. Although short term depletion of PSPC1 abolished Cajal body formation, prolonged depletion appeared to induce a hyperactivation of coilin expression, presumably as part of an adaptation response, with some clones presenting with telomerase Cajal body retention as seen with NONO and SFPQ loss. This suggests that PSPC1 can similarly facilitate telomerase

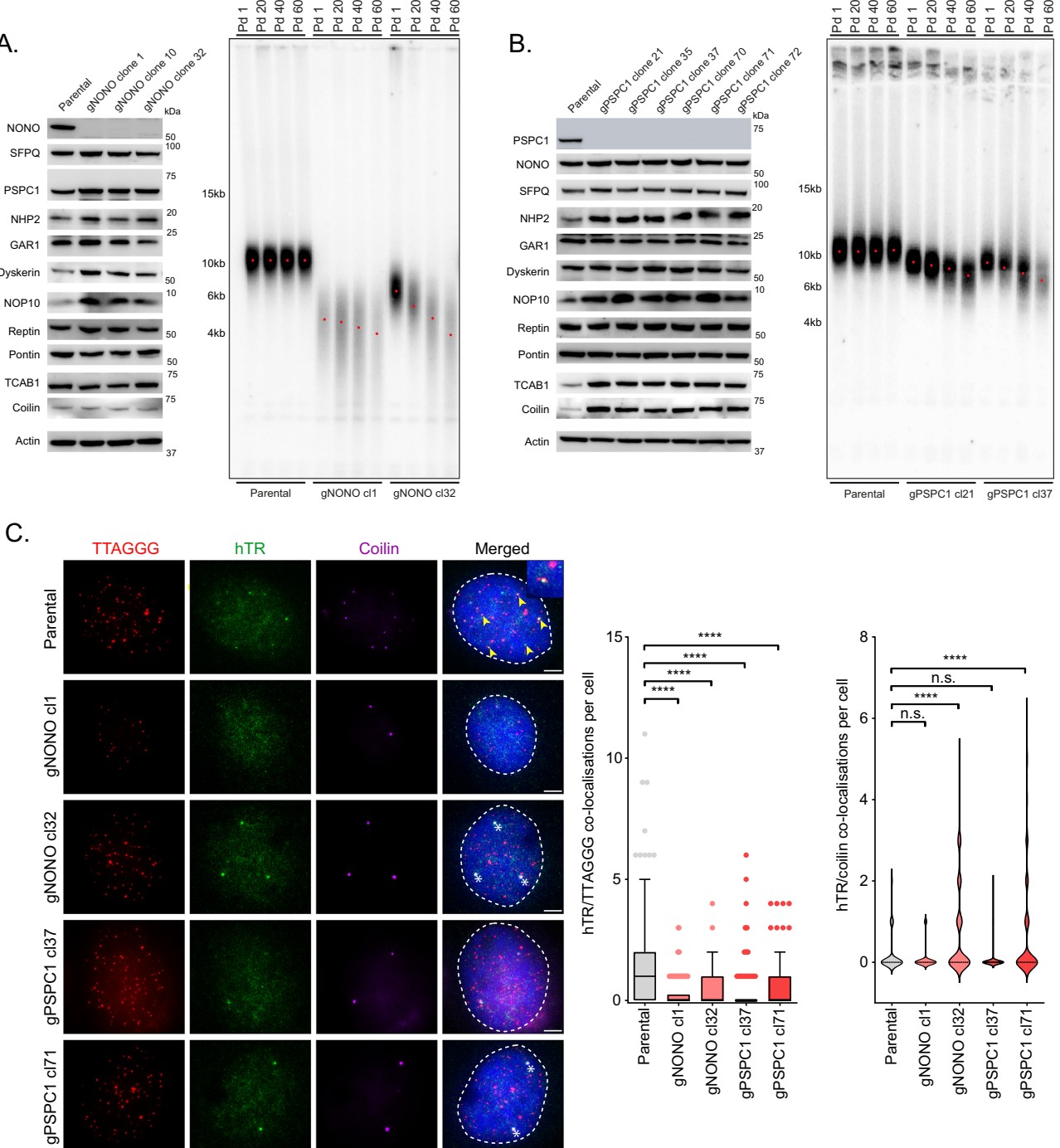

**Fig. 4 | Genetic disruption of NONO and PSPC1 causes telomere shortening and reduced telomerase recruitment in HT1080 cells. A** Western blot of DBHS and telomerase biology associated proteins in parental HT1080 and gNONO CRISPR edited clones (left panel). Telomere lengths measured by TRF Southern blot with cell passage (right panel). Pd, population doublings. Peak intensity of telomere length is indicated by red dot. **B** Western blot of DBHS and telomerase biology associated proteins in parental HT1080 and gPSPC1 CRISPR edited clones (left panel). Telomere lengths measured by TRF Southern blot with cell passage (right panel). Pd, population doublings. Peak intensity of telomere length is indicated by red dot. **C** Representative images of telomere (red), hTR (green), and coilin (purple) co-localizations in parental HT1080, gNONO, and gPSPC1 CRISPR edited clones (left panel). TTAGGG/hTR co-localizations are indicated by yellow arrows; hTR/coilin co-localizations are indicated by white asterisks. Scale bars are 5 µm. Tukey boxplots (median, interquartile range, and Tukey whiskers) of TTAGGG/hTR co-localizations (middle panel). Out of three experiments, $n$ = 150 cells scored per treatment, ****$p$ < 0.0001, Kruskal-Wallis test. Violin plots of hTR/coilin co-localizations (right panel). Out of three experiments, $n$ = 150 cells scored per treatment, n.s. = non-significant, ****$p$ < 0.0001, Kruskal-Wallis test. Source data are provided as a Source data file.

trafficking out of Cajal bodies. Telomerase localization to Cajal bodies is regulated by TCAB1, while the DBHS proteins promote release. However, DBHS protein binding to telomerase does not appear to be increased by TCAB1 or coilin depletion, precluding the possibility that these proteins compete for telomerase binding to prevent TCAB1 mediated sequestration in Cajal bodies.

The effects of NONO and SFPQ depletion on telomerase recruitment and retention in Cajal bodies is reminiscent of TPP1 and TIN2

depletion[10]. As TPP1 and TIN2 levels are not affected by DBHS depletion, this raises the possibility that the DBHS proteins may function independently to stabilise the association of telomerase at the telomere, as opposed to directly trafficking the telomerase RNP complex from Cajal bodies. However, as depleting one member of the DBHS family impairs telomerase recruitment without impacting the ability of the others to interact with hTERT, this suggests that the DBHS proteins interact with telomerase independently of telomere localisation.

Unlike the lack of functional redundancy in telomerase recruitment, the role of SFPQ in telomerase trafficking out of Cajal bodies can be substituted for by both NONO and PSPC1 overexpression. However, SFPQ overexpression cannot rescue telomerase trafficking in NONO depleted cells. In addition to providing further evidence for PSPC1 in promoting telomerase release from Cajal bodies, this one-way functional redundancy suggests that trafficking is not being restored solely due to replenishment of total DBHS protein levels. As only one member is being overexpressed, SFPQ rescue is most likely due to the formation of NONO and PSPC1 homodimers due to these individual proteins being in excess compared to their potential endogenous DBHS heterodimer partner. These homodimers differ from their heterodimer counterparts, including the presence of a well-ordered N-terminal β-clasp that promotes high-affinity RNA binding[27]. In contrast, SFPQ homodimers contain a short α-helix in this position that promotes polymerization and DNA binding[28]. We theorize that the NONO and PSPC1 overexpressed homodimers bind hTR with high affinity via the β-clasp to rescue hTR Cajal body sequestration, and that the SFPQ homodimer is not sufficient to perform this action. The propensity for the DBHS proteins to form heterodimers over homodimers endogenously may account for the lack of rescue by endogenous NONO and PSPC1 after SFPQ depletion[26,38]. Interestingly, PSPC1 overexpression cannot rescue telomerase Cajal body retention induced by NONO depletion, suggesting a regulator role for NONO beyond hTR binding. Despite our data suggesting that the DBHS proteins play a role in telomerase trafficking out of Cajal bodies, the inability for NONO and PSPC1 to promote telomerase recruitment after SFPQ depletion suggests that reduced recruitment is not caused solely by retention in Cajal bodies. This observation is more apparent in NONO and PSPC1 CRISPR knockout clones that have adapted and restored telomerase trafficking, but still experience telomere shortening due to reduced recruitment. This is most likely due to the observed reduced capacity of DBHS proteins to interact with the telomere when one member is depleted.

All four major constituents of the DBHS region appear to be essential for telomerase recruitment to the telomere. The observed dominant negative and lethal phenotypes associated with mutating the polymerization domain of both SFPQ and PSPC1 also indicate an essential role for this function in cancer cell survival. However, only the NONO and SFPQ DBHS regions appear to be individually essential for telomerase trafficking out of Cajal bodies. The RRM2 domain of PSPC1 was not essential for rescue. As the p.ΔDim mutant could not rescue trafficking, this would suggest that heterodimers formed with mutant p.ΔRRM2 PSPC1 may form a stable complex with other DBHS proteins capable of promoting telomerase release from Cajal bodies.

In addition to the broad spectrum of functions associated with the DBHS proteins in the literature, their expression and dimerization patterns appear to be cell type specific, with different dimers potentially having cell-type specific functions[12]. The DBHS proteins have also been shown to have opposing roles within disease subtypes, with NONO being shown to both promote and inhibit breast cancer proliferation[39–41]. The cell line specific effects of DBHS proteins on telomerase activity also fit this pattern. NONO and SFPQ promote hTR accumulation in HT1080, with prolonged depletion of NONO resulting in an upregulation of the other hTR stabilizing RNA binding proteins DKC1 and NOP10[5]. In contrast, NONO and SFPQ depletion resulted in the formation of more active telomerase complex in 293T. Although

they are not absolutely required for telomerase assembly, Cajal bodies have been hypothesized to be sites of telomerase RNP assembly, maturation, and accumulation[5,42]. As NONO and SFPQ depletion prevents telomerase trafficking out of Cajal bodies, this could result in a buildup of assembled telomerase complex within these nuclear bodies, which would account for the increased telomerase assembly and activity observed after transient depletion. This hypothesis accounts for the lack of change in telomerase activity observed with PSPC1 depletion, which does not cause telomerase Cajal body retention.

Impaired telomerase recruitment caused by PSPC1 depletion was consistently associated with telomere shortening. Long term depletion of NONO also caused telomere shortening associated with reduced telomerase recruitment in HT1080, HeLa, and HCT116 cells. However, long term depletion of NONO caused increased telomere lengthening without significantly impacting telomerase recruitment specifically in 293 cells. This lengthening was not dependent on SV40 T-antigen transfection or due to increased telomerase activity, changes in the DNA damage response, or activation of ALT phenotypes. These anomalous results highlight the complexity of telomere maintenance, as telomere lengthening was observed without altering traditional markers of telomere maintenance. Nevertheless, it is clear that disrupting members of the DBHS protein family consistently impact telomere length, and that PSPC1 is essential for telomere length maintenance by telomerase.

Overall, our results provide mechanistic insight into a previously unidentified role of the DBHS protein family in regulating telomerase biology (Fig. 5). All three members of the protein family regulate telomerase, with each member having distinct yet complementary roles that ensure correct telomerase recruitment. Our comprehensive approach has also revealed a role for the DBHS proteins in telomerase trafficking out of Cajal bodies, telomerase assembly, and hTR stability. These findings reveal several previously unidentified aspects of telomerase recruitment regulation and provide a potential strategy for manipulating telomerase function through DBHS protein modulation in cancer cells, which we have shown is feasible through the chemical inhibition of NONO.

## Methods
### Plasmids
Wild type DBHS proteins (NONO, SFPQ, PSPC1) and their functional mutants (ΔRRM1, ΔRRM2, ΔPol, ΔDim) were cloned into the pLenti-C-Myc-DDK-IRES-Puro backbone (Origene Technologies; PS100069) and/or the pLenti-C-Myc-DDK-IRES-Neo backbone (Origene Technologies; PS100081) by restriction enzyme subcloning of gene blocks synthesized by Integrated DNA Technologies (IDT). Wild type hTR and Δ1-20 hTR were cloned as a U3-hTR "cassette" into pcDNA3.1 expressing dyskerin as described previously[43]. NONO shRNA (shNONO B) in a pGFP-C-shLenti backbone (TL311143) was obtained from Origene Technologies. Wild type hTERT in a pLXSN backbone (Clontech; K1060) was a gift from Dr. Tony Cesare (Children's Medical Research Institute, University of Sydney). Lentivirus was produced by the Vector and Genome Engineering Facility (Children's Medical Research Institute, University of Sydney). For transient expression, cells were transfected at approximately 50% confluency with plasmid DNA using FuGENE®6 at a 3:1 ratio of FuGENE to DNA, according to manufacturer recommendations (Promega). For transient transfections plasmid DNA expression was allowed to proceed for 72 h prior to downstream analysis. For stable overexpression/knockdown, cells were transduced with lentivirus, allowed to recover for 24 h, and subject to G418 or puromycin selection. Cells were maintained under selection to ensure sustained overexpression/knockdown.

### Cell culture and cell lines
The cell lines HeLa (female), U-2 OS (female), HCT116 (male), HT1080 (male), 293 (female), and 293T (female) were obtained from the

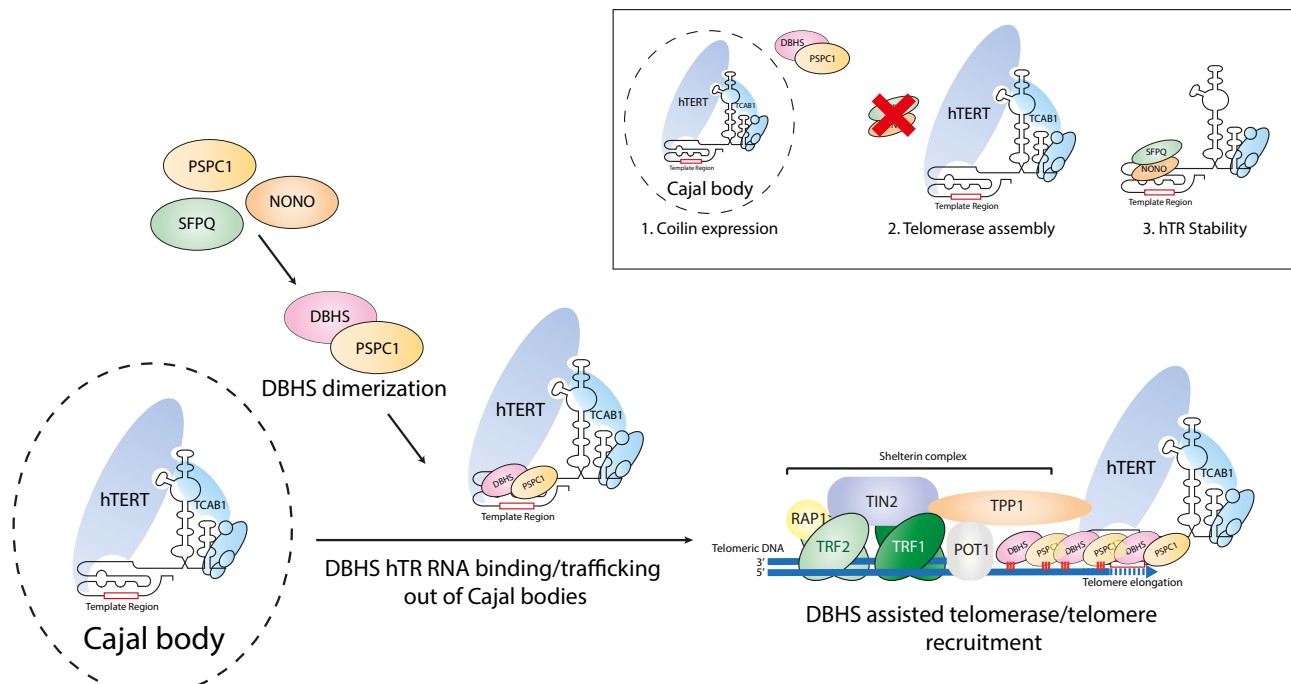

**Fig. 5 | Model for DBHS mediated telomerase recruitment to telomeres.** The DBHS proteins (NONO, SFPQ, PSPC1) dimerize with each other to facilitate telomerase trafficking out of Cajal bodies by interacting with the hTR telomerase subunit. The DBHS proteins then facilitate telomerase recruitment to the telomere, potentially stabilizing the interaction through polymerization. In addition to recruitment, PSPC1 is also involved in Cajal body formation through coilin expression (1), while NONO and SFPQ play cell line specific roles in reducing telomerase assembly with hTR (2) and hTR stability (3).

American Type Culture Collection. HeLa, U-2 OS, HCT116, HT1080, 293 and 293T were cultured in Dulbecco's modified Eagle's medium (DMEM) supplemented with 10% (v/v) fetal bovine serum (FBS) in a humidified incubator at 37 °C with 10% CO2. The following commercial compounds were used in cell treatments: dimethyl sulfoxide (DMSO, Sigma-Merck), Thymidine (Sigma-Merck), and Aphidicolin (Sigma-Merck). Cells were also treated with (R)-SKBG-1 and (S)-SKBG-1, which were generated in this study. Cell lines were authenticated by 16-locus short-tandem-repeat profiling and tested for mycoplasma contamination by CellBank Australia (Children's Medical Research Institute, University of Sydney).

### CRISPR knockout
NONO and PSPC1 knockout by CRISPR/Cas9 genome editing was performed commercially by the Vector and Genome Engineering Facility (VGEF) (Children's Medical Research Institute, University of Sydney) in HT1080, and 293T cells.

### RNA interference
The following Silencer and Silencer Select siRNAs were designed and synthesized by Life Technologies: NONO (s9614), SFPQ (s12712), PSPC1 (s30595), TCAB1 (s30251), Coilin (s15662), 3′UTR NONO (108251) and the Silencer Select RNAi siRNA Negative Control #2 (Scrambled; #4390847). Cell suspensions were transfected at 20–50% confluency with Lipofectamine RNAiMAX (Life Technologies) at a final siRNA concentration of 30 nM. Culture media was changed after 48 h and cells harvested for analysis 72 h post-transfection. Knockdown efficiency was validated by immunoblot analysis.

### Genomic DNA extraction and purification
Cells were harvested via trypsinization, washed in PBS, and resuspended in lysis buffer (50 mM Tris-HCl, 100 mM NaCl, 50 mM ethylenediamine tetra-acetic acid (EDTA), 0.5% SDS, pH8). Lysed cells were subjected to RNase A (50 μg/ml) treatment for 20 min at room temperature, followed by protein digestion with 400 μg/ml of proteinase K (Invitrogen) overnight at 55 °C. DNA was extracted using two rounds of phenol/chloroform extraction followed by ethanol precipitation.

### Terminal restriction fragment (TRF) analysis
Genomic DNA was digested with 4 U/μg of HinfI and RsaI overnight at 37 °C. Digested DNA was precipitated with 0.1 volume of 3 M sodium acetate pH 5.2 and 2.5 volumes of 100% ethanol. DNA was washed with 70% ethanol, dried, and dissolved in 10 mM Tris pH 8.4. Digested DNA (2 μg) was loaded on 1% (w/v) pulse-field certified agarose (Bio-Rad) gels and separated at 6 V/cm for 14 h at 14 °C with an initial switch time of 1 s and a final switch time of 6 s. The gels were dried for 75 min at 60 °C, denatured in 0.5 M NaOH/1.5 M NaCl for 1 h, and neutralized in 0.5 M Tris–HCl (pH 8.0)/1.5 M NaCl for 1 h. Gels were then rinsed in 2× SSC and pre-hybridized in Church buffer (250 mM sodium phosphate buffer, pH 7.2, 7% [wt/vol] SDS, 1% [wt/vol] BSA fraction V grade, and 1 mM EDTA) for 2 h at 50 °C. Gels were hybridized overnight with a γ-[32 P]-ATP-labeled (TTAGGG)₄ oligonucleotide probe at 50 °C, washed three times in 0.1× SSC for 15 min at 37 °C, and exposed to a PhosphorImager screen overnight. Imaging was performed on the Typhoon FLA 7000 system (GE Healthcare) with a PMT of 750 V.

### C-circle assay
C-circles were amplified with Phi29 polymerase using dATP, dTTP, and dGTP for 12 h at 37 °C. Products were dot blotted onto Biodyne B membranes (Pall) and pre-hybridized in PerfectHyb Plus (Sigma) for at least 30 min at 37 °C. γ-[32 P]-ATP-labeled telomeric C-probe (CCCTAA)₄ was then added and blots were hybridized overnight at 37 °C. Blots were washed with 0.2× SSC three times for 10 min each, then exposed to a PhosphorImager screen. Imaging was performed on the Typhoon FLA 7000 system (GE Healthcare) with a PMT of 750 V.

## Immunofluorescence and fluorescence in-situ hybridization (IF-FISH)

Cells on coverslips were subjected to pre-extraction by incubation in KCM permeabilization solution (120 mM KCl, 20 mM NaCl, 10 mM Tris, 0.1% (v/v) Triton X-100) for 10 min. Cells were then washed in phosphate buffered saline (PBS), fixed at room temperature for 10 min in PBS with 4% (v/v) formaldehyde and blocked with 100 µg/mL DNase-free RNase A (Sigma) in antibody-dilution buffer (20 mM Tris-HCl, pH 7.5, 2% (w/v) BSA, 0.2% (v/v) fish gelatin, 150 mM NaCl, 0.1% (v/v) Triton X-100 and 0.1% (w/v) sodium azide) for 1 h at room temperature. Cells were incubated with primary antibody diluted in antibody-dilution buffer for 1 h at room temperature, washed in PBS Tween-20, and incubated with secondary antibody diluted in antibody-dilution buffer for 1 h at room temperature. Coverslips were rinsed with PBS then fixed with 4% (v/v) formaldehyde at room temperature prior to telomere FISH. Coverslips were subjected to a graded ethanol series (75% for 2 min, 85% for 2 min, and 100% for 2 min) and allowed to air-dry. Dehydrated coverslips were overlaid with 0.3 µg/ml TAMRA−OO-(CCCTAA)$_3$ or 0.3 µg/mL Alexa 488−OO-(CCCTAA)$_3$ telomeric PNA probe in PNA hybridization solution (70% deionized formamide, 0.25% (v/v) NEN blocking reagent (PerkinElmer), 10 mM Tris−HCl, pH 7.5, 4 mM Na2PO4, 0.5 mM citric acid, and 1.25 mM MgCl2), or *NEAT1*_m-Q570 probe (LGC Biosearch Technologies) denatured at 80 °C for 3 min, and hybridized at room temperature overnight. Coverslips were washed twice with PNA wash A (70% formamide, 10 mM Tris pH 7.5) and then PNA wash B (50 mM Tris pH 7.5, 150 mM NaCl, 0.8% Tween-20) for 5 min each. DAPI was added at 50 ng/ml to the second PNA wash B. Finally, coverslips were rinsed briefly in deionized water, air dried and mounted in DABCO (2.3% 1,4 Diazabicyclo (2.2.2) octane, 90% glycerol, 50 mM Tris pH 8.0) or ProLong™ Gold Antifade Mountant (Thermofisher Scientific). Microscopy images were acquired on a Zeiss Axio Imager microscope with appropriate filter sets.

## EdU detection

Cells were pulsed with 10 µM EdU for 2 h. Cells were permeabilized in KCM solution, then fixed with 4% formaldehyde PBS solution. The Click-iT® Alexa Fluor 647 azide reaction was then performed according to the manufacturer's instructions, before blocking with antibody-dilution buffer and RNaseA.

## Automated image analysis

ZEN microscopy images (.czi) were processed into extended projections of z-stacks and imported into Cellprofiler v4.2.1[44] for analysis. The DAPI channel was used to mask individual nuclei as primary objects. Foci within each segmented nucleus were identified using an intensity-threshold based mask. Any given object was considered overlapping another object when at least 80% of the first object's area was enclosed within the area of a second object.

## Western blot

Cells were collected and lysed in RIPA buffer (50 mM Tris-HCl pH 7.6, 150 mM NaCl, 1% Nonidet P-40, 0.5% sodium deoxycholate, 0.1% SDS, 4 mM EDTA) supplemented with cOmplete Mini EDTA-free protease inhibitor cocktail (Roche). Proteins were resolved on a 4%–12% bis-tris gel (Invitrogen). Transferred membranes were blocked in 5% milk and incubated with primary antibody overnight at 4 °C. Membranes were then incubated with corresponding HRP-conjugated secondary antibodies (Dako) for 1 h at room temperature, and bands visualized using PICO enhanced chemiluminescence reagents (Thermo Scientific). Uncropped and unprocessed scans with accompanying molecular weight/size markers of all blots presented in this study are available in the Source Data file.

## Immunoprecipitation (IP)

Cells were lysed in buffer A (20 mM HEPES-KOH pH 8, 0.3 M KCl, 2 mM MgCl$_2$, 0.1% (v/v) Triton X-100, 10% (v/v) glycerol) supplemented with 1 mM PMSF (Cell Signaling) for 1 h at 4 °C. Following lysis, the lysate was cleared at 16,000 × g for 30 min. Supernatant was then mixed with 5–15 µg of antibody and 20 µL Protein G/agarose beads overnight at 4 °C. Telomerase immunoprecipitation was performed with αhTERT prepared as described previously[43]. The beads were then washed with 10 mL buffer A using vacuum suction. Proteins were then eluted under either non-denaturing or denaturing conditions. For non-denaturing conditions, proteins were eluted in Buffer A supplemented with 1 mM DTT and 10 µg/ml peptide antigen solution (c-Myc: EQKLISEEDL; hTERT: ARPAEEATSLEGALSGTRH) for 30 min. For denaturing conditions, proteins were eluted in 100 µl buffer A supplemented with 1 mM PMSF and 1× NuPAGE reducing agent (Invitrogen) for 10 min at 80 °C.

## RNA dot blot

RNA dot blot was performed on hTR isolated in buffer A or TERRA and *COIL* mRNA isolated using the RNeasy mini kit (Qiagen)[43]. Isolated RNA was diluted 1:10 v/v in formamide buffer (90% v/v Deionised formamide; 10% v/v 10× TBE (900 mM tris base; 900 mM boric acid; 10 mM EDTA); 0.01% w/v xylene cyanol; 0.01% w/v bromophenol blue) and denatured at 80 °C for 10 min. RNA solutions were then dot blotted onto Biodyne B membranes (Pall) and pre-hybridized in PerfectHyb Plus (Sigma) for 1 h at 55 °C. γ-[$^{32}$P]-ATP-labeled probe (hTR: CGGTGGAAGGCGGCAGGCCGAGGC; TERRA: (CCCTAA)$_4$; *COIL* mRNA: GGAGAGGTCATACTCAGGCAG) was then added and blots hybridized overnight at 55 °C. Blots were washed with 0.2× SSC three times for 10 min each, then exposed to a PhosphorImager screen. Imaging was performed on the Typhoon FLA 7000 system (GE Healthcare) with a PMT of 750 V.

## In vitro transcription of hTR

Human telomerase RNA component (hTR) was transcribed in vitro using T7 RNA polymerase in the presence of α−$^{32}$P-GTP to produce radiolabelled RNA for downstream electrophoretic mobility shift assays. Plasmid containing wild-type hTR was linearised with FokI restriction enzyme at the 3′ end of the hTR gene to ensure efficient termination of transcription. 50 µg/mL linearised pUC-hTR DNA template was combined with 50 mM Tris−HCl (pH8.0), 10 mM MgCl$_2$, 2 mM spermidine, 10 mM dithiothreitol (DTT), 1 mM each of ATP, CTP, GTP, and UTP (TriLink Biotechnologies), 5% (v/v) inorganic pyrophosphatase (New England Biolabs) and approximately 100 µCi of α-32P-GTP in nuclease-free water. The reaction was initiated by addition of 1 µg/mL T7 RNA polymerase (New England Biolabs) and incubating at 37 °C for 2 h.

The reaction was quenched with addition of 20 mM EDTA followed by brief vortexing. To remove unincorporated nucleotides, the reaction mixture was purified using a NICK size-exclusion column (17085502, Cytiva) pre-equilibrated with TE buffer (10 mM Tris−HCl, 1 mM EDTA, pH8.0). The sample was loaded onto the column and the radiolabelled RNA was eluted with TE buffer. The radiolabelled RNA was further purified with two extractions with phenol/chloroform/isoamyl alcohol (P2069, Merck). The aqueous layer containing purified, radiolabelled hTR was aliquoted and stored at −80 °C until required.

## Electrophoretic mobility shift assay (EMSA)

Purified protein was diluted to desired concentration, ranging from 0- to 10-fold, in buffer A (20 mM HEPES-KOH pH 8, 0.3 M KCl, 2 mM MgCl$_2$, 0.1% (v/v) Triton X-100, 10% (v/v) glycerol) and kept on ice while preparing EMSA master mix. EMSA binding buffer (20 mM HEPES-KOH, pH7.9, 100 mM KCl, 0.8 mM MgCl$_2$, 0.2 mM EDTA, 5% (v/v) glycerol) was supplemented with 100 ng/mL recombinant albumin (B9200S, New England Biolabs), 2 mM PMSF (8553 Cell Signalling

technology), 2 mM DTT and 100 ng/mL tRNA (R8508, Merck) fresh on the day and mixed thoroughly. To this, 0.5 μL per sample of in vitro transcribed, radiolabelled hTR RNA was added and mixed thoroughly to produce 2X EMSA master mix. To each diluted protein sample, an equal volume of EMSA master mix was added, mixed gently, and incubated at RT for 45 min. After incubation, 5X TBE sample loading buffer (LC6678, ThermoFisher Scientific) was added to each reaction to a final concentration of 0.5X and the samples were mixed thoroughly. Samples were loaded into a 0.8% agarose, 0.5X TBE gel and electrophoresed at 150 V for 135 min at 4 °C. The gel was subsequently dried at 80 °C for 35 min, briefly washed in dH$_2$O, sealed and exposed to a phosphor screen (Cytiva) overnight. The screens were scanned the following day with Amersham Typhoon imager (Cytiva) at 4000 intensity.

## Telomerase Repeat Amplification Protocol (TRAP)
TRAP was performed on non-denatured IP products according to Kim and Wu, 1997[45] with modifications. IP products were incubated with M2 primer (AATCCGTCGAGCAGAGTT) for 30 min at 37 °C and denatured at 90 °C for 20 min. Telomeric DNA was then amplified using 100 ng M2 and 50 ng ACX primers (GCGCGGCTTACCCTTACCCTTACCC-TAACC) in a Quick-Load® Taq 2X Master Mix (New England Biolabs) according to manufactures instructions. TRAP products were detected by Syber Gold staining (Thermofisher Scientific) after resolution in a 10% polyacrylamide gel with 0.5× tris–borate–EDTA buffer.

## Quantitative PCR TRAP (qTRAP)
Cells were collected and lysed in buffer A (20 mM HEPES-KOH pH 8, 0.3 M KCl, 2 mM MgCl$_2$, 0.1% (v/v) Triton X-100, 10% (v/v) glycerol) supplemented with 1 mM PMSF (Cell Signaling) for 1 h at 4 °C. Following lysis, the lysate was cleared at 16,000 × g for 30 min. Cell lysates containing 5 μg protein were incubated with M2 primer (AATCCGTC-GAGCAGAGTT) for 30 min at 37 °C and denatured at 90 °C for 20 min. Telomeric Quantitative PCR was then performed using 100 ng M2 and 50 ng ACX primers (GCGCGGCTTACCCTTACCCTTACCCTAACC) in SensiFAST™ Direct Probe SuperMix (Bioline) for 40 amplification cycles according to the manufacturer's instructions. Heat denatured lysate was used as a negative control for each sample.

## Quantitative reverse transcription PCR (qPCR)
Total RNA was isolated using the RNeasy mini kit (Qiagen) and DNase-treated to remove genomic DNA. Reverse transcription was performed with 2 μg of isolated RNA using SuperScript™ III Reverse Transcriptase (Thermofisher Scientific) according to the manufacturer's instructions. Quantitative PCR was performed using SensiFAST™ Direct Probe SuperMix (Bioline) according to the manufacturer's instructions. Coilin primers (F:AAATGGAGCCGAGGTAGTGG, R: GGAGAGGTCA-TACTCAGGCAG) and GAPDH primers (F: ACCCACTCCTCCACCTTTG, R: CTCTTGTGCTCTTGCTGGG) were run with an annealing temperature of 60 °C. PCRs were performed on cDNA equivalent to 100 ng of total RNA and carried out for 40 amplification cycles, followed by melt curve analysis. For each sample, a replicate omitting the reverse transcription step was included as a negative control. Real-time data were analysed using the ΔΔC(t) method, with GAPDH as the reference gene, and expressed as fold change relative to the appropriate control (±SEM).

## Telomere chromatin immunoprecipitation (ChIP)
Subconfluent cells (1 × 10$^7$) were harvested by trypsinization and resuspended in 1 mL cell lysis buffer (5 mM HEPES-KOH, pH 8.0, 85 mM KCl, 0.4% (v/v) NP-40, 1× CPI, and 1 mM PMSF). Cells were incubated on ice for 10 min and centrifuged at 6000 g for 15 s at 4 °C to pellet nuclei. After removal of the supernatant, nuclei were fixed at room temperature in the same volume of cell lysis buffer lacking NP-40 and containing 1% (v/v) formaldehyde. Fixed nuclei were incubated at room

temperature for 10 min with mixing once by inversion during fixation. Cross-linking was quenched by addition of 75 μL of 1.5 M glycine and incubation for 5 min at room temperature, with two inversions during incubation. The mixture was centrifuged again at 6000 g for 15 s at 4 °C and the supernatant discarded. To the nuclei, 800 μL of ice-cold buffer A (20 mM HEPES-KOH pH 8, 0.3 M KCl, 2 mM MgCl$_2$, 0.1% (v/v) Triton X-100, 10% (v/v) glycerol) supplemented with 1 mM PMSF (Cell Signaling) was added and incubated for 20 min at 4 °C. Chromatin was fragmented with a Bioruptor Twin Sonication Device (Diagenode) for 8 × 60 s pulses on high setting with a 60 s pause and was centrifuged at 17,000 g for 10 min at 4 °C to remove the solubilized chromatin within the supernatant. For the immunoprecipitation, 100 μL of chromatin was added to 400 μL of ice-cold buffer A supplemented with 1 mM PMSF (Cell Signaling), and mixed with 5–15 μg of antibody and 20 μL Protein G/agarose beads overnight at 4 °C. The beads were then washed with 10 mL buffer A using vacuum suction. Proteins were eluted in 100 μl buffer A supplemented with 1 mM PMSF and 1x NuPAGE reducing agent (Invitrogen) for 10 min at 80 °C. Isolated chromatin was diluted 1:10 v/v in formamide buffer (90% v/v Deionised formamide; 10% v/v 10 × TBE (900 mM tris base; 900 mM boric acid; 10 mM EDTA); 0.01% w/v xylene cyanol; 0.01% w/v bromophenol blue) and denatured at 80 °C for 10 min. Chromatin solutions were then dot blotted onto Biodyne B membranes (Pall) and pre-hybridized in PerfectHyb Plus (Sigma) for 1 h at 55 °C. γ-[32 P]-ATP-labeled probe (telomere: (TTAGGG)$_4$; Alu: CGCCCGGCTAATTTTTGTAT) was then added and blots hybridized overnight at 55 °C. Blots were washed with 0.2× SSC three times for 10 min each, then exposed to a PhosphorImager screen. Imaging was performed on the Typhoon FLA 7000 system (GE Healthcare) with a PMT of 750 V.

## IF-FISH (hTR/TTAGGG/coilin)
Cells grown on coverslips were treated with 2 mM thymidine for 15 h, released for 9 h, and blocked again with 0.5 μg/mL aphidicolin for 15 h to arrest cells in early S phase. Cells were then released for 3 h into mid S-phase (confirmed through EdU detection), fixed in 2% (vol/vol) paraformaldehyde in PBS for 15 min and permeabilised in 0.5% IGEPAL/PBS for 10 min. Cells were then rinsed in Milli-Q H$_2$O and further permeabilised/dehydrated in 1:1 acetone-methanol for 10 min. Cells were then washed in PBS and blocked in antibody-dilution buffer (20 mM Tris-HCl, pH 7.5, 2% (w/v) BSA, 0.2% (v/v) fish gelatin, 150 mM NaCl, 0.1% (v/v) Triton X-100 and 0.1% (w/v) sodium azide) for 1 h at room temperature. Cells were incubated with αCoilin (Santa Cruz; # sc-55594) diluted 1:500 in antibody-dilution buffer for 1 h at room temperature, washed in PBS Tween-20, and incubated with secondary antibody diluted in antibody-dilution buffer for 1 hr at room temperature. Coverslips were rinsed with PBS then fixed with 4% (v/v) formaldehyde at room temperature prior to telomere FISH. Coverslips were subjected to a graded ethanol series (75% for 2 min, 85% for 2 min, and 100% for 2 min) and allowed to air-dry. Dehydrated coverslips were overlaid with 5 ng each of hTR oligonucleotide probe mix (hTR 1: 5′-Cy5- GCTGACATTTTTTGTTTGCTCTAGAATGAACGGTGGAAGGCG GCAGGCCGAGGCTT-3′; hTR 4: 5′-Cy5- CTCCGTTCCTCTTCCTGCG GCCTGAAAGGCCTGAACCTCGCCCTCGCCCCCGAGAG-3′; hTR 5: 5′-Cy5- ATGTGTGAGCCGAGTCCTGGGTGCACGTCCCACAGCTCAGGGA ATCGCGCCGCGCGC -3′) and telomere probe (5′-TxRed-CCCTAA CCCTAACCCTAACCCTAACCCTAACCCTAACCCTAACCCTAA CCCTAACCCTAACCCTAA-3′) in fluorescent in situ hybridization (FISH) buffer (10% dextran sulfate, 2 mM vanadyl-ribonucleoside complex, 0.02% RNase-free BSA, 1 μg/μl Escherichia coli tRNA [Roche], 2× SSC [1× SSC is 0.15 M NaCl plus 0.015 M sodium citrate], 50% formamide), denatured at 80 °C for 3 min, and hybridized at 37 °C overnight. Coverslips were washed twice with FISH wash 1 (2× SSC pH 7.2/50% formamide/0.1% SDS) and then FISH wash 2 (4× SSC pH 7.2/ 0.1% Tween) for 10 min each. Finally, coverslips were rinsed briefly in deionized water, air dried and mounted in ProLong™ Gold Antifade

with DAPI (Thermofisher Scientific). Microscopy images were acquired on a Zeiss Axio Imager microscope with appropriate filter sets.

## Synthesis of (R)-SKBG-1 and (S)-SKBG-1
Chemical synthesis and validation for (R)-SKBG-1 and (S)-SKBG-1 are described in Supplementary Methods.

## Antibodies and antibody validation
The antibodies used throughout this study, including dilutions used for experiments, can be found in Supplementary Table 1. DBHS antibodies used for ChIP were validated through siRNA depletion experiments (Supplementary Fig. 5F).

## Quantification and statistical analysis
Statistical analysis was performed using GraphPad Prism. Box plots are displayed using the Tukey method where the box extends from the 25th to the 75th percentile data points and the line represents the median. The upper whisker represents data points ranging up to the 75th percentile+ (1.5 × the inner quartile range), or the largest value data point if no data points are outside this range. The lower whisker represents data points ranging down to the 25th percentile− (1.5 × the inner quartile range), or the smallest data point if no data points are outside this range. Data points outside these ranges are shown as individual points. Violin plots are displayed using the extended method, whereby the width of the violin is directly related to the estimated distribution of the data at a given Y value. Error bars, statistical methods and n are described in figure legends.

## Reporting summary
Further information on research design is available in the Nature Portfolio Reporting Summary linked to this article.

## Data availability
All data supporting the findings of this study are available within the article and its Supplementary Information files. Data are also available from the corresponding author upon request. Source data are provided with this paper.

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

## Acknowledgements

The authors thank Josh Stern and Melissa Kartawinata (Bryan laboratory) for optimizing the FISH procedure for detection of telomerase at telomeres. The author(s) acknowledge support from Luminesce Alliance. The Alliance is comprised of five partners Sydney Children's Hospitals Network, the Children's Medical Research Institute, the Children's Cancer Institute, the University of Sydney, and the University of New South Wales Sydney. Microscopy was performed in the ACRF Telomere Analysis Center, supported by the Australian Cancer Research Foundation and the Ian Potter Foundation. This work was funded by the National Health and Medical Research Council of Australia (2003250 to H.A.P. and Y.H.L.), Tour De Cure (RSP-262-2020 to A.P.S.), and Cancer Institute NSW (ECF171269 to A.P.S.).

## Author contributions

A.P.S. conceptualized the idea. A.P.S. and H.A.P. supervised the project. A.P.S. designed and performed most experiments. X.W. was responsible for the synthesis of (R)-SKBG-1 and (S)-SKBG-1. Y.H.L. planned and supervised the synthesis of (R)-SKBG-1 and (S)-SKBG-1. J.K.W., M.C. and C.B.N. performed experiments. A.P.S. and H.A.P. wrote the manuscript. S.B.C., T.M.B., C.B.N. and A.F. evaluated experiments and provided editorial input.

## Competing interests

The authors declare no competing interests.
