## [Transparent Peer Review file · Nature Communications]

NONO, SFPQ, and PSPC1 promote telomerase recruitment to the telomere

Corresponding Author: Professor Hilda Pickett

Version 0:

Reviewer comments:

Reviewer #1

(Remarks to the Author)

Reviewer comment:

In the current manuscript Sobinoff et al. describe an interesting connection between the DBHS proteins SFPQ/NONO/PSPC1 and the human telomerase complex. Based on public available data that connect these proteins with telomere length the authors perform elegant experiments with the goal to elaborate a model where DBHS proteins interact with the telomerase complex to modulate the trafficking between Cajal bodies and telomeres. This pathway is proposed to impact on telomere length regulation. However, a significant part of the experimental result is not reproducible in the different cell models used in this study. This is most probably due to the different origin of the cell models and also due to the wide aspects of molecular processes that are modulated by individual DBHS proteins. In addition, DBHS proteins were also shown to be involved in phase separation, further increasing the complexity of the framework of this study. To my opinion this renders it very difficult to fit all proteins into a common mechanism. I also appreciate the effort the authors use to provide a detailed and objective description of obtained phenotypes and aiming to convert this into a functional model. Nevertheless, the cell-type to cell-type variation of obtained phenotypes is an obstacle to the definition of a functionally validated pathway that controls telomere length homeostasis in a DBHS protein dependent manner. In addition, a de-regulation of the proposed pathway in disease or development has not been presented. For this reason, I cannot support the publication of the current version of the manuscript.

Abstract:

Authors mention in line 21 that loss of DBHS proteins result in telomere shortening. This is not sustained by all the experiments on telomere length shown in the manuscript.

DBHS proteins interact with active telomerase via hTR.

The association of DBHS proteins is novel, very interesting and holds biological relevance. However, more experiments should be done to really confirm observations and relevance. Experiments should be performed with all DBHS proteins to prove the claim of the title of the paragraph. Additional, independent experimental approaches, including in vitro experiments should be performed.

General comments:

The human telomerase complex was purified in the past, is there evidence that DBHS proteins co-purified with telomerase. If not, what could be the reason?

Loading controls for protein extracts used in IP experiments and Northern dot blots should be shown in respective figure panels (and all other figures). Otherwise it is difficult to sustain the conclusions of the manuscript.

Authors should highlight whether endogenous proteins were immunoprecipitated. Data should be reproducible using endogenous or ectopically expressed target proteins.

Specific comments:

sFig1A: authors should show the individual data (DBHS expression/telomere length) related to the cell lines used in this study (HT1080, 293T, 293, HeLa and HCT116)

Fig 1A-B:

Authors should validate the specificity of DBHS protein interaction with hTR in more detail.

- Northern dot should be hybridized with probes for additional RNAs such as hnRNAs, NEAT, tRNAs, TERRA or similar to verify the selectivity of DBHS interaction with hTR. This is also necessary because DBHS proteins are highly expressed, increasing the risk for non specific interaction with RNAs (in the cell lysate).
- A competition experiment with in vitro transcribed hTR or un-related control RNAs would be highly informative
- To sustain the claim of interaction with hTR, in vitro binding studies with recombinant proteins and in-vitro transcribed hTR should be provided. This would also sustain the claim of hTR – DBHS protein interaction
- RIP experiments would be useful

Figure 1C:

The TRAP experiments are very nice. It would be important to do IP-TRAP for other DBHS proteins. Does the assay also work using endogenously expressed NONO/PSPC1/SFPQ?

Figure 1D. Elegant. But are same results obtained when doing SFPQ or PSPC1 IPs? Loading controls are missing (especially in the RNA dot blot).

Figure 1E. Does ectopic expression of hTR or the addition in vitro transcribed hTR reconstitute the interaction of hTERT with DBHS proteins?

Fig. 1F. A classic experiment to connect Cajal bodies with telomeres and telomerase. DBHS proteins appear to promote the association of hTR to telomeres. Given the central importance of this statement, a validation using an independent method would be required (ChIP based? PLA?).

Given that the work centers on DBHS proteins, the localization of these proteins with respect to Cajal bodies and telomeres should be characterized, taking also in consideration the accumulation of SFPQ/NONO/PSPC1 in paraspeckles.

Authors should also validate “Cajal body features” in DBHS loss of function conditions – for example, Cajal body number and size; presence of TCAB1 or other Cajal body components that have a link to telomerase function.

Additional questions:

- Related to Fig. 1F: What about the impact of SFPQ, NONO or PSPC1 on hTR and hTERT? That may be mentioned already here. What about the integrity/stability of hTR under these conditions?
- Related to Figure 1F. the particular hTR and Coilin staining pattern in siPSPC1 cells should be discussed.

Cell line specific effects of DBHS proteins on telomerase assembly.

Supplementary figure 2 does not provide a clear information on the role of DBHS protein on telomerase assembly. Effects appear to be specific and different in the cancer cell line used. Why is that? If DBHS proteins are central for telomerase regulation, effects after knock-down should be reproducible in all cell lines used. Does it eventually relate to higher basal DBHS protein expression levels in one cell line, compared to the other? Do DBHS accumulate stronger on paraspeckles in one line versus the other line. This part of the manuscript is confusing to read.

Additional comment:

- In sFig2A the anti-hTERT WB is missing for the Input.
- Line 135: here the authors explain to use affinity purification, later they mention immunity purification. Please clarify.

NONO/PSPC1 overexpression rescues telomerase trafficking in SFPQ depleted cells

In this part of the manuscript the authors show that all DBSH proteins have a role in recruitment of telomerase/hTR to telomeres (LOF phenotypes cannot be rescued by ectopic expression of other DBSH proteins). The results on localization of hTR to Cajal bodies in rescue experiments are not linear and do not give the possibility to make a rational conclusion. With regards to the stable cells lines: at this point of the manuscript I am missing experiments that show that ectopic (and stable) expression of epitope-tagged NONO rescues effects caused by transient depletion of endogenous NONO (same would be needed for SFPQ and PSPC1). In the context of depletion of an endogenous DBHS protein the ectopically expressed protein should not induce alterations in telomerase recruitment/trafficking. I think that this is shown in Fig. 3. In this case, please anticipate the information + discuss eventual dominant negative effects of ectopically expressed proteins. In Fig2F hTERT WB is missing for the input.

Authors observe that individual DBSH proteins bind to hTERT independently from each other; in contrast there appears to be some co-operation between DBSH proteins in binding telomere repeats. I am not sure how this observation can help to understand or define a molecular mechanism for the dynamics of hTR/telomerase dynamics between Cajal bodies and telomeres.

I assume that the authors want to claim that the telomerase complex “migrates” together with SFPQ/NONO/(PSPC1) to the telomeres. However, it can also be expected that SFPQ/NONO/PSPC1 may already be loaded at the telomeres and is waiting to “receive” and stabilize the telomerase complex at chromosome ends. I think a detailed examination of SFPQ/NONO/PSPC1 localization in the relevant nuclear compartments in the presence or absence of hTR/hTERT can be very helpful to define a mechanism.

A functional DBSH region is crucial for telomerase recruitment

In this part of the manuscript the authors did a really remarkable effort in generating model cells to understand the role of individual protein domains in hTR retention in Cajal bodies and hTR recruitment to telomeres. To me, all the results are difficult to interpret, presumably also because of the heterogeneity of basic phenotypes observed in 293 and HT1080 cells after SFPQ/NONO/PSPC1 depletion (precisely described by the authors in sFig.2). In addition, overexpression constructs for mutant DBHS proteins appear to mediate dominant negative effects. Maybe it might be better to make conclusions only considering dominant negative effects resulting from ectopic expression of DBHS mutants (keeping endogenous DBHS proteins at normal levels???)

Does mutant DBHS proteins still interact with hTR and hTERT (in IP and IP-RNA dot blot)

In the current version of the text I agree with the authors (line 232) that the large amount of data can give “suggestions” but not a strong conclusion.

Chemical inhibition of NONO impairs telomerase recruitment in cells

Nice data confirming the relevance of NONO. Experiments have been done in 293T cells. What about in HT1080 cells? It would also be nice to get an idea on the cytotoxicity of compound concentrations used in the experiments. The representative images of the IF are not clear – to me the signals for hTR and Cajal bodies change in the different experimental conditions. It would also be interesting to get an idea on the impact of the compounds on Cajal body size and number/nucleus. Does NONO inhibition impact on the association of NONO/SFPQ/PSPC1 with hTR and/or telomere DNA repeats? The amount of data (all supplementary data) is quite small for a separate paragraph.

NONO and PSPC1 depletion causes cancer cell specific changes in telomere length

The authors have generated an impressive number of 293T/HT1080 loss of function models using CRISPR/KO and studied telomere length and hTR trafficking. This is a very essential part of the study as it addresses physiological relevance of DBHS proteins for long-term telomere length homeostasis, with possible impact on cancer and aging.

Data on PSPC1 is consistent in both cell models: telomere shortening. No SFPQ CRISPR lines were obtained and long term, knock-down of SFPQ was not done/not possible.

Although high expression of NONO has been anticipated to be linked with long telomeres in cancer cell lines (sFig 1A) the situation is unclear in the used experimental cell models. Considering CRISPR and shRNA experiments, loss of NONO can result telomere elongation, shortening or no change. The impact of long term treatment with NONO inhibitors is not known. Unfortunately, this part of the manuscript does not allow to clearly connect mechanistic findings of previous figures into a long-term physiological relevance. How was the correlation of telomere length and DBHS expression for HT1080, 293T, 293, HeLa and HCT116 cells in the original analysis shown in sFig1A?

General comments to immunofluorescent images.

Quality should be improved and co-localization should be shown by zooming into the relevant sections.

Reviewer #2

(Remarks to the Author)

I co-reviewed this manuscript with one of the reviewers who provided the listed reports.

Reviewer #3

(Remarks to the Author)

The manuscript by Sobinoff et al. investigates the role of DBHS family RNA/DNA binding proteins in regulating telomerase trafficking through Cajal bodies to telomeres. High expression levels of DBHS proteins NONO, SFPQ, and PSPC1 were found to be associated with cell lines possessing long telomeres. Co-immunoprecipitation (Co-IP) of DBHS proteins revealed their interaction with telomerase RNP in cultured human cells. Transient depletion of DBHS increased Cajal body retention and reduced telomeric recruitment of telomerase, as well as a cell line-dependent effect on TR accumulation and telomerase assembly. Rescue experiments suggest that DBHS proteins have overlapping functions in promoting telomerase trafficking out of Cajal bodies, but not in telomere recruitment. Inhibition of NONO with the small molecule compound C145 also impaired telomerase recruitment but had no effects on telomerase trafficking. Importantly, long-term depletion of NONO and PSPC1 in HT1080 cells resulted in progressive telomere shortening. However, in 293T cells NONO and PSPC1 knockout exhibited telomere elongation and shortening phenotypes, respectively.

Overall, this paper is well written and brings forth an intriguing new concept of telomerase regulation. However, due to the complexity of the subjects involved, the conclusions are not supported by firm experimental evidence. The main challenge arises from inconsistent results between studies of transient and long-term depletion of DBHS proteins, and among different cell types. The lack of biochemical characterization of the interactions between DBHS proteins and telomerase makes the

proposed model for DBHS in telomerase regulation speculative.

1. The investigation into the interaction between DBHS proteins and hTR was based on the immunoprecipitation of endogenous or ectopically expressed NONO protein. Since DBHS can form homo- or hetero-dimers, when hTR is detected by Co-IP with NONO (Fig. 1A & 1D), it suggests that NONO-containing dimers, such as NONO-NONO, NONO-PSPC1, and NONO-SFPQ, may interact with hTR. However, this experiment is not sufficient to conclude that ALL three DBHS proteins interact with hTR (Lines 86, 98). Therefore, the statement that DBHS proteins interact with active telomerase via hTR (Lines 68, 103) does not supported by the data.

2. Given that DBHS proteins can form homo- and hetero-dimers, depletion of one DBHS protein may prevent the formation of multiple dimers, leading to functional defects attributable to the loss of these dimers. Although it might not be technically feasible to examine the functions of all DBHS dimers, the authors should compare the levels of DBHS proteins in the cell lines used. This comparison may help explain the cell type-dependent roles of DBHS in telomerase assembly and telomere length regulation.

3. The study involving chemical inhibition of NONO for telomerase recruitment analysis was performed in 293T cells (Fig. S5). To illustrate the broader application of this strategy, the chemicals should be applied to both 293T and HT1080 cells for a telomere length study. Additionally, it is crucial to determine if the chemicals affect Coilin and hTR levels.

4. The authors contend that long-term NONO depletion impairs telomerase recruitment and causes telomere shortening (Line 302), based on studies in HT1080, HeLa, and HCT116 cells. However, the studies on HeLa and HCT116 cells are limited to TRF assays. To support this claim, the effects of NONO depletion on telomerase recruitment and Cajal body retention in HeLa and HCT116 cells should be examined.

5. In ChIP experiments, non-specific antibody binding can lead to false positive results. To demonstrate antibody specificity, a sample with depletion of the target protein should be included in the ChIP experiments. Unfortunately, this control was excluded from the ChIP experiments shown in Fig. 2E. Be aware that using IgG as a control is not a suitable replacement.

Minor Issues:

1. My experience is that ectopically expressing TERT in U2OS cells is sufficient to restore their telomerase activity, suggesting the presence of hTR in U2OS cells. Perhaps the U2OS strain used in this study could be indeed negative for both hTR and hTERT. The authors should provide real-time RT-PCR data to support this.

2. The sentences starting from Line 208 and Line 210 seem to contradict each other.

3. Due to the lack of evidence for a direct physical interaction between DBHS proteins and hTR, the model should be modified.

Reviewer #4

(Remarks to the Author)

In this manuscript, Sobinoff and colleagues report a novel role for the RNA-binding proteins NONO, SFPQ and PSPC1, which are part of the DBHS family, in telomerase trafficking and recruitment at telomeres. The authors provide evidence that members of the DBHS family are associated with the catalytically active telomerase and with telomeres. Furthermore, in the absence of DBHS proteins, telomerase RNA accumulates in Cajal bodies, and the recruitment of telomerase at telomeres is disrupted, leading to telomeres shortening. The authors propose that DBHS proteins are novel components of the telomerase trafficking machinery.

Overall, the authors have not convincingly showed that the effects observed on telomerase recruitment and telomeres length upon depletion of NONO, SFPQ or PSPC1 are direct and are not due to indirect effects, as these proteins play wide roles in RNA metabolism, including splicing and translation. For instance, the authors have not looked at the impact of NONO, SFPQ and PSPC1 depletion on the expression of the Shelterin proteins, especially TPP1 and TIN2, which promote telomerase recruitment at telomeres. More important, several of the phenotypes observed upon depletion of DBHS proteins (accumulation of hTR in Cajal bodies, decreased recruitment of hTR at telomeres, telomeres shortening), have been observed in TPP1 or TIN2 mutants (Abreu et al., MCB, 2010; Zhong et al., Cell, 2012). The impact of DBHS protein depletion on the expression and alternative splicing of these proteins should be determined.

More important, there are several contradictory results and inconsistencies presented in this manuscript. The impact of NONO, SFPQ and PSPC1 depletion on telomerase activity, recruitment or telomeres length not only varies among cell lines but also depends on the type of depletion used (siRNAs versus CRISPR KO), and varies among CRISPR clones (see comments below). This raises concerns regarding the specific effect of DBHS proteins function in telomerase activity at telomeres.

Finally, the claim that DBHS proteins are involved in telomerase trafficking is not supported by the data. As mentioned above, disruption of the TPP1-hTERT interaction could explain several of the results. The authors do not provide evidence for a direct role for DBHS proteins in telomerase RNA trafficking.

Major comments:

- Figure 1A: following an IP of NONO, the authors claim that all three proteins (NONO, SFPQ and PSPC1) interact with hTR. Since NONO interacts with the two other proteins independently of hTR, this experiment does not support their claim. The authors have not performed co-IPs with PSPC1 or SFPQ to detect their interaction with hTR.

- Figure 2: The authors should check if the association of the DBHS proteins at telomeres is RNA-dependent or not. These proteins are known to bind TERRA and RNA:DNA hybrids at telomeres (Petti et al., Nature Communication, 2019). RNase-treatment prior the ChIP assay would help answer this question.

- The results reported are confusing. While depletion of NONO, SFPQ or PSCP1 by siRNA decreases hTR/telomeres colocalization in 293T cells (Figure 1G), deletion of NONO with Cas9 in 293T cells does not affect hTR/telomeres colocalization (Figure S6B) and results in longer telomeres in these cells (Figure S6C and D).

Figure S6E: The two CRISPR clones of 293T deleted of NONO (clone 5 and clone 11) have opposite effect on telomerase activity (in one clone it goes up and in the other clone it goes down). Both clones show the same level of recruitment of hTR at telomeres, similar to the control cells. However, both clones show telomere elongation over time! These results are contradictory.

- Telomerase activity using qTRAP should be performed on the CRISPR clones of NONO and PSCP1 to determine if these KO affect telomerase activity.

Figure S2B: hTR levels are normal in 293T cells depleted of NONO, SFPQ or PSCP1, but they are low in HT1080 cells with the same depletions.

Figure S2E: depletion of NONO and SFPQ using siRNAs increases telomerase activity in 293T cell, but decreases telomerase activity in HT1080 cells

- In 293T cells treated with siRNA against PSCP1, hTR levels and formation of Cajal bodies is very low. Why?

- Figure S5: Have you measured hTR levels in cells treated with the SKBG-1 drug? Have you validated the activity of the drug on the formation of NONO aggregates? How do you explain that the drug does not increase the accumulation of hTR in Cajal bodies, as the siNONO does?

Minor comments:

- Figure 1C: why NONO needs to be overexpressed in cells for the qTRAP assay?

- Legend of Figure 1: The legend mention that hTR was detected by Northern blot. However, dot blots are presented in this figure. This should be corrected.

- Figure 2: ChIP should be reported as % of input. The current labeling of the Y axis "Telomeric DNA normalized to input (AU)" is confusing.

Version 1:

Reviewer comments:

Reviewer #1

(Remarks to the Author)

The authors have made a significant effort to improve the original version of the manuscript. In particular, the new EMSA experiments are highly informative (note: the legend of Figure 1 lacks information on EMSAs). The authors have responded clearly to the majority of comments. The new title of the manuscript is more appropriate.

I support publication, although the contradictory results on telomere length in 293 and 293T cells after long-term NONO loss of function experiments do not support the proposed model. I appreciate the ALT validation in these long-term models; however, the manuscript does not provide a clear explanation for these contradictory results.

Therefore, I suggest that the statement in line 445 of the discussion: "...these anomalous results highlight the complexity of telomere maintenance" should be improved to provide a better explanation or hypothesis.

A comment on the abstract: the statement "...culminating in progressive telomere shortening in several cell lines" is misleading. It implies that telomeres shorten in the several cell lines tested; the statement does not clarify that in some cell lines the telomere length phenotype is different. This information should be included in the abstract, giving credit to the complex interplay of the DBHS proteins.

Reviewer #2

(Remarks to the Author)

Reviewer #3

(Remarks to the Author)

The revised manuscript provides substantial evidence supporting the interaction between DBHS proteins and telomerase. The study presents novel findings demonstrating that DBHS proteins directly regulate intracellular telomerase function, influencing telomere length homeostasis. However, the varying effects of DBHS protein loss-of-function on telomere lengthening across different cell types suggest potential cell-type-specific roles for individual DBHS proteins, and thus the underlying mechanisms require further investigation.

Reviewer #4

(Remarks to the Author)

In this revised manuscript, the authors provide new data regarding the role of DBHS proteins in telomeres biology. Overall, they have properly answered several of the comments and questions raised by the reviewers. This reviewer agrees with the authors saying, "What is clear, is that disrupting members of the DBHS protein family consistently impacts telomere length, and that PSPC1 is essential for telomere length maintenance by telomerase". Moreover, the authors now provide convincing evidence for DBHS proteins interaction with telomerase.

However, there is still no clear mechanism, and I am not convinced that the telomere phenotypes are due to a role of DBHS proteins in recruitment of telomerase to the telomeres, as the title of the manuscript suggests.

The authors put a lot of emphasis on the role of DBHS proteins in telomerase recruitment, but their data also point toward a role for these proteins in hTR expression and telomerase activity, especially in HT1080 cells (sFigure 4).

In figure 4, the authors have not measured hTR levels and telomerase activity in the NONO and PSPC1 KO HT1080 cells. Telomeres shortening in these KO cells may well be explained by reduced hTR levels. This may also explain the reduction in hTR/telomeres colocalization event per cell, as lower level of hTR would decrease the number of hTR/telomeres colocalization events. hTR levels have not been quantified in HeLa and HCT116 cells depleted of NONO using shRNAs, so it is not clear that changes in hTR expression is a cell line specific effect, as claimed by the authors.

In general, quantification of hTR RNA level should be done more thoroughly in the manuscript. hTR RNA was quantified using dot blots, which cannot distinguish between hTR size variants (mature versus unprocessed). Furthermore, no loading control was included in the input of the dot blots (like Actin mRNA or U1 snRNA) in sFigure 4A-B, so it is not clear how accurately hTR level is measured with these blots. Northern blot or RT-qPCR would be more appropriate.

Version 2:

Reviewer comments:

Reviewer #4

(Remarks to the Author)

By including their new data, the authors have properly answered my comments in this revised version of their manuscript. They also clarified some comments in the text. Although there are still questions regarding the cell line specific effect of the DBHS proteins on telomere homeostasis, I support the publication of this manuscript in this journal.

RESPONSE TO REVIEWERS

Referee #1:

Reviewer #1: In the current manuscript Sobinoff et al. describe an interesting connection between the DBHS proteins SFPQ/NONO/PSPC1 and the human telomerase complex.

Based on public available data that connect these proteins with telomere length the authors perform elegant experiments with the goal to elaborate a model where DBHS proteins interact with the telomerase complex to modulate the trafficking between Cajal bodies and telomeres. This pathway is proposed to impact on telomere length regulation. However, a significant part of the experimental result is not reproducible in the different cell models used in this study. This is most probably due to the different origin of the cell models and also due to the wide aspects of molecular processes that are modulated by individual DBHS proteins. In addition, DBHS proteins were also shown to be involved in phase separation, further increasing the complexity of the framework of this study. To my opinion this renders it very difficult fit all proteins into a common mechanism. I also appreciate the effort the authors use to provide a detailed and objective description of obtained phenotypes and aiming to convert this into a functional model.

Nevertheless, the cell-type to cell-type variation of obtained phenotypes obstacle the definition of a functionally validated pathway that controls telomere length homeostasis in a DBHS protein dependent manner. In addition, a de-regulation of the proposed pathway in disease or development has not been presented. For this reason, I cannot support the publication of the current version of the manuscript.

Abstract:

Authors mention in line 21 that loss of DBHS proteins result telomere shortening. This is not sustained by all the experiments on telomere length shown in the manuscript.

Loss of PSPC1 caused telomere shortening in all cell lines tested (Figure 4B; sFigure 8A), while loss of NONO caused telomere shortening in all cell lines apart from 293 and 293T (Figure 4A; sFigure 8C;8D). The abstract has been amended to clarify these observations.

DBHS proteins interact with active telomerase via hTR.

The association of DBHS proteins is novel, very interesting and hold biological relevance. However, more experiments should be done to really confirm observations and relevance. Experiments should be performed with all DBHS proteins to proof the claim of the title of the paragraph. Additional, independent experimental approaches, including in vitro experiments should be performed.

This is an important point raised by multiple reviewers. Immunoprecipitation of hTERT showed that all three DBHS proteins interact with the catalytic component of telomerase (Figure 1C), and that this association is abolished in the absence of hTR (Figure 1F). We have addressed this point using three different approaches. First, we performed RIP experiments with SFPQ and PSPC1 (in addition to NONO) to further demonstrate the direct association between all three DBHS proteins and hTR (Figure 1A). Second, we performed EMSAs with recombinant DBHS proteins to confirm hTR binding in vitro (Figure 1B). Third, we demonstrated that re-introduction of hTR restores the interaction between the DBHS proteins and telomerase (Figure 1F). These data are now included in Figure 1 of the revised manuscript.

General comments:

The human telomerase complex was purified in the past, is there evidence that DBHS proteins co-purified with telomerase. If not, what could be the reason?

The DBHS proteins have not previously been identified to co-purify with telomerase, highlighting the novelty of our study. It is worth noting that other important and verified interactors, such as TCAB1, were also not identified in the original mass spec analyses that identified dyskerin, NOP10, NHP2 and GAR1 as being associated with the telomerase complex. In fact, TCAB1 was first identified as a telomerase associated protein through the isolation of dyskerin complexes¹. The previous lack of co-purification can be explained by limitations in the telomerase purification and protein detection techniques.

Loading controls for protein extracts used in IP experiments and Northern dot blots should be shown in respective figure panels (and all other figures). Otherwise it is difficult to sustain the conclusions of the manuscript.

Loading controls, i.e. immunoprecipitated protein/RNA and input are shown for all experiments².

The exception to this is immunopurification of endogenous hTERT, where the input is absent. This is because it is not possible to detect endogenous hTERT via western blot without immunopurification, as previously described by the Cech group in 2014³. Other groups that examine immunopurified endogenous hTERT do not include an input control for the same reasons⁴. In the instances where we have looked at endogenous hTERT levels, we employed an established hTERT immunoprecipitation (IP) protocol in which endogenous hTERT is immunoprecipitated from cell lysates using a sheep polyclonal antibody, and then eluted with the corresponding peptide antigen^{3, 5}.

Authors should highlight whether endogenous proteins were immunoprecipitated. Data should be reproducible using endogenous or ectopically expressed target proteins.

It is stated in the text and figure legends when ectopically expressed target proteins were immunopurified. In all other instances endogenous proteins were used for purification.

All interactions with overexpressed components (Figure 1E-F) were also shown to interact with endogenous components (Figure 1A; Figure 1C). The exception to this is the IP-TRAP experiments, which were only performed using overexpressed DBHS proteins (Figure 1D; sFigure 1E). This is because IP-TRAP requires the purification of telomerase associated proteins under non-denaturing conditions to preserve enzymatic activity. We utilised an anti-Myc tag antibody to IP exogenously expressed c-Myc tagged DBHS proteins, which allowed us to competitively elute DBHS complexes with c-Myc peptide under non-denaturing conditions, thereby preserving telomerase complex activity.

Specific comments:

sFig1A: authors should show the individual data (DBHS expression/telomere length) related to the cell lines used in this study (HT1080, 293T, 293, HeLa and HCT116).

Expression levels of the DBHS proteins via western blot and telomere length via TRF analysis for HT1080, 293T, 293, HeLa and HCT116 have now been included as sFigure 1B and sFigure 1C, respectively.

Fig 1A-B:

Authors should validate the specificity of DBHS protein interaction with hTR in more detail.

- Northern dot should be hybridized with probes for additional RNAs such as hnRNAs, NEAT, tRNAs, TERRA or similar to verify the selectivity of DBHS interaction with hTR. This is also necessary because DBHS proteins are highly expressed, increasing the risk for non-specific interaction with RNAs.

Like other telomerase associated RNA binding proteins (dyskerin, NOP10, NHP2 and GAR1) the DBHS proteins have been shown to bind to several different RNAs, including NEAT1 and TERRA. Our study demonstrates that all three DBHS proteins interact directly with hTR. We are not suggesting that this interaction is exclusive; however, we do demonstrate that PSPC1 does not interact with coilin RNA in 293T and HT1080 cell lysates, indicative of selectivity for hTR (sFigure 1L).

- A competition experiment with in vitro transcribed hTR or un-related control RNAs would be highly informative.

- To sustain the claim of interaction with hTR, in vitro binding studies with recombinant proteins and in-vitro transcribed hTR should be provided. This would also sustain the claim of hTR – DBHS protein interaction.

We have performed EMSAs with recombinant DBHS proteins (NONO, SFPQ, and PSPC1) and *in-vitro* transcribed hTR, demonstrating concentration-dependent binding between the DBHS proteins and hTR. A super-shift was also observed in the presence of antibodies against immunopurified DBHS proteins, supporting the specificity of their binding to hTR. These data are now included in Figure 1B.

- RIP experiments would be useful.

RNA immunoprecipitation (RIP) experiments involve immunoprecipitation of the RNA binding protein of interest together with its associated RNA for identification of bound transcripts. The immunoprecipitation experiments in which hTR was found to associate with immunopurified DBHS proteins are RIP experiments, and comprehensively demonstrate interaction between the DBHS proteins and hTR (Figure 1A; 1E-F).

Figure 1C:

The TRAP experiments are very nice. It would be important to do IP-TRAP for other DBHS proteins. Does the assay also work using endogenously expressed NONO/PSPC1/SFPQ?

We have now performed IP-TRAP for SFPQ and PSPC1 in 293T cells, demonstrating their association with active telomerase complex. These data are now included in sFigure 1E.

IP-TRAP requires the purification of telomerase associated proteins under non-denaturing conditions to preserve enzymatic activity. We do not have a way of extracting endogenous DBHS associated proteins under non-denaturing conditions.

Figure 1D. Elegant. But are same results obtained when doing SFPQ or PSPC1 IPs? Loading controls are missing (especially in the RNA dot blot).

We have performed these experiments, and similar results were obtained with SFPQ and PSPC1 IPs. These data are included in Figure 1E of the revised manuscript.

Loading controls, i.e. immunoprecipitated protein/RNA and input are shown for all experiments². The western blot and RNA dot blot samples were obtained from the same eluate.

Figure 1E. Does ectopic expression of hTR or the addition in vitro transcribed hTR reconstitute the interaction of hTERT with DBHS proteins?

This is a great suggestion. Ectopic expression of hTR does reconstitute the interaction between DBHS proteins and hTERT in U-2 OS cells. These data have been included as Figure 1F.

Fig. 1F. A classic experiment to connect Cajal bodies with telomeres and telomerase. DBHS proteins appear to promote the association of hTR to telomeres. Given the central importance of this statement, a validation using an independent method would be required (ChIP based? PLA?).

hTR/TTAGGG FISH is the gold standard in the field for examining telomerase recruitment and stands alone when assaying endogenous telomerase recruitment. Alternative methods (ChIP, PLA) are challenging due to 1) the low expression of hTERT in cells, and 2) the absence of a suitable hTERT antibody for IF experiments. To address this comment, we attempted hTERT ChIP in DBHS depleted cells, but were unable to get it to work consistently with several antibodies.

We feel that the observations of reduced telomerase recruitment associated with decreased telomere length throughout this study in several cell lines, coupled with the magnitude of the reduction observed with DBHS protein depletion, strongly support the claims made.

Given that the work centers on DBHS proteins, the localization of these proteins with respect to Cajal bodies and telomeres should be characterized, taking also in consideration the accumulation of SFPQ/NONO/PSPC1 in paraspeckles.

We (ChIP; Figure 2E) and others⁶ have shown that both NONO and SFPQ interact with telomeres. We have now also included data showing PSPC1 association with telomeres (ChIP; Figure 2E). In addition, we have now performed IF-FISH experiments demonstrating DBHS/TTAGGG and DBHS/coilin co-localizations, which are included in sFigure 2A-C of the revised manuscript.

To determine the potential role for paraspeckles in DBHS/telomerase recruitment/biology, we have also looked at NEAT1/hTR and NEAT1/TTAGGG co-localisations, and found hTR to be rarely associated with paraspeckles (sFigure 2D). Paraspeckles were more frequently associated with telomeres, although this was not specific to telomerase positive cells (sFigure 2E).

These data are now included and discussed in the manuscript.

Authors should also validate “Cajal body features” in DBHS loss of function conditions – for example, Cajal body number and size; presence of TCAB1 or other Cajal body components that have a link to telomerase function.

We have now examined the effects of DBHS protein depletion on Cajal body number and Cajal body association with TCAB1 in 293T and HT1080 cells. Consistent with our western blot analysis (sFigure 1I), PSPC1 depletion caused a noticeable decrease in Cajal body foci number (sFigure 1K). However, no reduction in Cajal body association with TCAB1 was observed when normalised to Cajal body number (sFigure 3B-C).

Additional questions:

- Related to Fig. 1F: What about the impact of SFPQ, NONO or PSPC1 on hTR and hTERT? That may be mentioned already here. What about the integrity/stability of hTR under these conditions?

The impact of DBHS protein depletion on hTR and hTERT levels, as well as telomerase assembly, is detailed in sFigure 4. Overall, the DBHS proteins do not influence hTERT expression, and cause a cell line specific effect on the accumulation of hTR and telomerase ribonucleoprotein assembly.

- Related to Figure 1F. the particular hTR and Coilin staining pattern in siPSPC1 cells should be discussed.

We have now quantified coilin foci in DBHS depleted cells (sFigure 1K) and expanded the discussion on the effects of siPSPC1 knockdown on coilin levels and coilin foci.

Cell line specific effects of DBHS proteins on telomerase assembly.

Supplementary figure 2 does not provide a clear information on the role of DBHS protein on telomerase assembly. Effects appear to be specific and different in the cancer cell line used. Why is that?

Transient NONO and SFPQ protein depletion result in some cell line specific variability in hTR stability, which in turn impacts telomerase ribonucleoprotein assembly and total telomerase activity via q-TRAP (sFigure 4). We have been unable to determine the precise nature of this variability. Nevertheless, the effects of DBHS protein depletion on telomerase recruitment, which is the major focus of this manuscript, are consistent and more striking (Figure 1H-I; sFigure 1G-H).

If DBHS proteins are central for telomerase regulation, effects after knock-down should be reproducible in all cell lines used. Does it eventually relate to higher basal DBHS protein expression levels in one cell line, compared to the other? Do DBHS accumulate stronger on paraspeckles in one line versus the other line. This part of the manuscript is confusing to read.

To address this comment, we performed western blot analysis and paraspeckle visualisation. DBHS protein expression levels were relatively stable (sFigure 1B), and paraspeckle formation was comparable across all cell lines examined (sFigure 2E).

Additional comment:

- In sFig2A the anti-hTERT WB is missing for the Input.

The input is missing because hTERT levels can only be determined through hTERT IP. A more detailed explanation is provided in a previous response.

- Line 135: here the authors explain to use affinity purification, later they mention immunity purification. Please clarify.

All purification was antibody dependent and therefore was immunopurified. We have clarified this throughout the manuscript.

NONO/PSPC1 overexpression rescues telomerase trafficking in SFPQ depleted cells

In this part of the manuscript the authors show that all DBSH proteins have a role in recruitment of telomerase/hTR to telomeres (LOF phenotypes cannot be rescued by ectopic

expression of other DBSH proteins). The results on localization of hTR to Cajal bodies in rescue experiments are not linear and do not give the possibility to make a rational conclusion. With regards to the stable cell lines: at this point of the manuscript I am missing experiments that show that ectopic (and stable) expression of epitope-tagged NONO rescues effects caused by transient depletion of endogenous NONO (same would be needed for SFPQ and PSCP1). In the context of depletion of an endogenous DBSH protein the ectopically expressed protein should not induce alterations in telomerase recruitment/trafficking. I think that this is shown in Fig. 3. In this case, please anticipate the information + discuss eventual dominant negative effects of ectopically expressed proteins.

The reviewer is correct, these data are shown in Figure 3. Stable expression of epitope-tagged NONO, SFPQ, or PSCP1 was able to rescue the effects caused by transient depletion of endogenous NONO, SFPQ, or PSCP1. Specifically, depletion of endogenous DBSH proteins in the presence of ectopically expressed protein did not induce alterations in telomerase recruitment/trafficking. This has now been clarified in the revised manuscript. Potential dominant negative effects of ectopically expressed DBSH mutants are also now mentioned in the results section.

In Fig2F hTERT WB is missing for the input.

The input is missing because hTERT levels can only be determined through hTERT IP. A more detailed explanation is provided in a previous response.

Authors observe that individual DBSH proteins bind to hTERT independently from each other; in contrast there appears to be some co-operation between DBSH proteins in binding telomere repeats. I am not sure how this observation can help to understand or define a molecular mechanism for the dynamics of hTR/telomerase dynamics between Cajal bodies and telomeres.

Our results show that depletion of one member of the DBSH protein family does not affect the ability of the remaining members to interact with hTERT (Figure 2F). In the absence of one DBSH protein family member, the remaining members can still form dimers with each other. To assay independence, we would have to deplete two members at a time out of the three. Unfortunately, cells do not grow under these conditions.

In contrast, depletion of one DBSH protein member does impact the ability of the remaining two to interact with the telomere (Figure 2E). This suggests that the presence of all three DBSH proteins is required for their optimal telomere association, but not for their association with telomerase, potentially explaining why overexpressing NONO and PSCP1 can compensate for the loss of SFPQ in telomerase trafficking out of Cajal bodies, but not in telomerase recruitment to the telomere. This rationale has been clarified in the revised manuscript.

I assume that the authors want to claim that the telomerase complex “migrates” together with SFPQ/NONO/(PSCP1) to the telomeres. However, it can also be expected that SFPQ/NONO/PSCP1 may already be loaded at the telomeres and is waiting to “receive” and stabilize the telomerase complex at chromosome ends. I think a detailed examination of SFPQ/NONO/PSCP1 localization in the relevant nuclear compartments in the presence or absence of hTR/hTERT can be very helpful to define a mechanism.

DBSH proteins “receiving” telomerase at the telomere, as opposed to directly facilitating telomerase recruitment to the telomere, is an interesting hypothesis. Indeed, depletion of

TIN2 or TPP1 causes hTR retention in Cajal bodies, although these results are complicated as recruitment was done in the presence of overexpressed telomerase⁷. However, the ability of NONO and PSPC1 overexpression to rescue hTR retention in Cajal bodies caused by SFPQ depletion, but not telomerase recruitment to the telomere, suggests that they are interacting with telomerase away from the telomere (Figure 2B-C). This is also supported by depletion of individual DBHS proteins preventing telomerase recruitment to the telomere, but not impacting the ability of the remaining DBHS proteins to interact with hTERT (Figure 2E-F). Further discussion of these points has now been included in the revised manuscript.

A functional DBSH region is crucial for telomerase recruitment

In this part of the manuscript the authors did a really remarkable effort in generating model cells to understand the role of individual protein domains in hTR retention in Cajal bodies and hTR recruitment to telomeres. To me, all the results are difficult to interpret, presumably also because of the heterogeneity of basic phenotypes observed in 293 and HT1080 cells after SFPQ/NONO/PSPC1 depletion (precisely described by the authors in sFig.2). In addition, overexpression constructs for mutant DBHS proteins appear to mediate dominant negative effects. Maybe it might be better to make conclusions only considering dominant negative effects resulting from ectopic expression of DBHS mutants (keeping endogenous DBHS proteins at normal levels???)

We appreciate that this comprehensive analysis is difficult to explain succinctly; however, we felt that, due to the heterodimerisation of the DBHS proteins, it was necessary to combine the mutant overexpression with endogenous knockdown. We have been as clear and concise as possible in our interpretation that some mutant constructs had a dominant negative effect on recruitment, and all mutant constructs were unable to rescue telomerase recruitment phenotypes caused by endogenous DBHS depletion. Both observations support the primary conclusion that the DBHS domain is important for telomerase recruitment.

Does mutant DBHS proteins still interact with hTR and hTERT (in IP and IP-RNA dot blot).

These experiments are difficult to design as the DBHS proteins function as heterodimers and oligomers. Therefore, mutant DBHS proteins will form dimers with endogenous DBHS proteins, complicating conclusions. For example, p.ΔRRM1, which shouldn't be able to bind RNA, may still be associated with hTR through its interaction with endogenous NONO, SFPQ, or PSPC1. In addition, n.ΔDim, which can't dimerise, may still be associated with hTR through its interaction with endogenous NONO, SFPQ, or PSPC1 through its polymerisation domain.

We have performed these experiments for NONO functional domain mutants in 293T and HT1080 cells (n=1). The mutants appear to have lower affinity for hTR, but the above considerations make interpretation difficult, and we do not think it is suitable to include in the manuscript. These data are included here for the reviewers' consideration.

In the current version of the text I agree with the authors (line 232) that the large amount of data can give “suggestions” but not a strong conclusion.

Chemical inhibition of NONO impairs telomerase recruitment in cells

Nice data confirming the relevance of NONO. Experiments have been done in 293T cells. What about in HT1080 cells? It would also be nice to get an idea on the cytotoxicity of compound concentrations used in the experiments. The representative images of the IF are not clear – to me the signals for hTR and Cajal bodies change in the different experimental conditions. It would also be interesting to get an idea on the impact of the compounds on Cajal body size and number/nucleus. Does NONO inhibition impact on the association of NONO/SFPQ/PSPC1 with hTR and/or telomere DNA repeats? The amount of data (all supplementary data) is quite small for a separate paragraph.

The experiments with the NONO inhibitor (R)-SKBG-1 were performed as a proof of concept that DBHS proteins can be chemically inhibited to reduce telomerase recruitment.

(R)-SKBG-1 cytotoxicity has been explored previously⁸. We have now performed the telomerase recruitment experiments with the inhibitor in HT1080 cells (sFigure 7B-C). In addition, we have also assayed Cajal body number, coilin integrated intensity per cell, and hTR integrated intensity per cell after treatment with (R)-SKBG-1 in 293T and HT1080 cells (sFigure 7D-F). Treatment with the inhibitor reduces telomerase recruitment in both cell lines, but does not appear to acutely alter Cajal body number and hTR levels. These data are now included in the revised manuscript.

NONO and PSPC1 depletion causes cancer cell specific changes in telomere length

The authors have generated an impressive number of 293T/HT1080 loss of function models using CRISPR/KO and studied telomere length and hTR trafficking. This is a very essential part of the study as it addresses physiological relevance of DBHS proteins for long-term telomere length homeostasis, with possible impact on cancer and aging.

Data on PSPC1 is consistent in both cell models: telomere shortening. No SFPQ CRISPR lines were obtained and long term, knock-down of SFPQ was not done/not possible.

Although high expression of NONO has been anticipated to be linked with long telomeres in cancer cell lines (sFig 1A) the situation is unclear in the used experimental cell models. Considering CRISPR and shRNA experiments, loss of NONO can result telomere elongation, shortening or no change. The impact of long-term treatment with NONO inhibitors is not known.

Unfortunately, this part of the manuscript does not allow to clearly connect mechanistic findings of previous figures into a long-term physiological relevance. How was the correlation of telomere length and DBHS expression for HT1080, 293T, 293, HeLa and HCT116 cells in the original analysis shown in sFig1A?

This is good point. In order to connect the telomere length changes more comprehensively to the initial analysis in sFigure 1A and thereby improve the flow of the manuscript, we have now included expression levels of the DBHS proteins via western blot and telomere length via TRF analysis for HT1080, 293T, 293, HeLa and HCT116 as sFigure 1B and sFigure 1C, respectively. NONO and SFPQ expression was consistent across all cell lines tested. PSPC1 expression varied between cell lines, but did not trend with telomere length.

NONO depletion caused long term telomere shortening associated with impaired telomerase recruitment in all cell lines tested (Figure 4A; 4C; sFigure 8D-E), with the exception of 293 and derived 293T cells (sFigure 8C; 8D). Indeed, telomere lengthening occurred in these cell lines via a mechanism we cannot explain even after extensive investigation (telomerase activity, TERRA levels, DNA damage, ALT phenotypes; sFigure 9). Intriguingly the 293 and 293T cell lines had the shortest baseline telomere lengths of all the cell lines analysed, although they are not drastically different to HCT116 (sFigure 1C). These anomalous results highlight the complexity of telomere length maintenance. What is clear, is that disrupting members of the DBHS protein family consistently impacts telomere length, and that PSPC1 is essential for telomere length maintenance by telomerase. These points have now been addressed in the discussion.

General comments to immunofluorescent images.

Quality should be improved and co-localization should be shown by zooming into the relevant sections.

We have now included zoomed-in sections highlighting TTAGGG/hTR co-localisations in all main figures.

Referee #2:

Reviewer #2: I co-reviewed this manuscript with one of the reviewers who provided the listed reports.

Referee #3:

Reviewer #3: The manuscript by Sobinoff et al. investigates the role of DBHS family RNA/DNA binding proteins in regulating telomerase trafficking through Cajal bodies to telomeres. High expression levels of DBHS proteins NONO, SFPQ, and PSPC1 were found to be associated with cell lines possessing long telomeres. Co-immunoprecipitation (Co-IP) of DBHS proteins revealed their interaction with telomerase RNP in cultured human cells. Transient depletion of DBHS increased Cajal body retention and reduced telomeric recruitment of telomerase, as well as a cell line-dependent effect on TR accumulation and telomerase assembly. Rescue experiments suggest that DBHS proteins have overlapping functions in promoting telomerase trafficking out of Cajal bodies, but not in telomere recruitment. Inhibition of NONO with the small molecule compound C145 also impaired telomerase recruitment but had no effects on telomerase trafficking. Importantly, long-term depletion of NONO and PSPC1 in HT1080 cells resulted in progressive telomere

shortening. However, in 293T cells NONO and PSPC1 knockout exhibited telomere elongation and shortening phenotypes, respectively.

Overall, this paper is well written and brings forth an intriguing new concept of telomerase regulation. However, due to the complexity of the subjects involved, the conclusions are not supported by firm experimental evidence. The main challenge arises from inconsistent results between studies of transient and long-term depletion of DBHS proteins, and among different cell types. The lack of biochemical characterization of the interactions between DBHS proteins and telomerase makes the proposed model for DBHS in telomerase regulation speculative.

1. The investigation into the interaction between DBHS proteins and hTR was based on the immunoprecipitation of endogenous or ectopically expressed NONO protein. Since DBHS can form homo- or hetero-dimers, when hTR is detected by Co-IP with NONO (Fig. 1A & 1D), it suggests that NONO-containing dimers, such as NONO-NONO, NONO-PSPC1, and NONO-SFPQ, may interact with hTR. However, this experiment is not sufficient to conclude that ALL three DBHS proteins interact with hTR (Lines 86, 98). Therefore, the statement that DBHS proteins interact with active telomerase via hTR (Lines 68, 103) does not supported by the data.

We have now performed Co-IP experiments with all DBHS proteins confirming their interaction with hTR and active telomerase. These data are included in Figure 1A; 1E, and sFigure 1E of the revised manuscript.

2. Given that DBHS proteins can form homo- and hetero-dimers, depletion of one DBHS protein may prevent the formation of multiple dimers, leading to functional defects attributable to the loss of these dimers. Although it might not be technically feasible to examine the functions of all DBHS dimers, the authors should compare the levels of DBHS proteins in the cell lines used. This comparison may help explain the cell type-dependent roles of DBHS in telomerase assembly and telomere length regulation.

Expression levels of the DBHS proteins via western blot and telomere length via TRF analysis for HT1080, 293T, 293, HeLa and HCT116 have now been included as sFigure 1B and sFigure 1C respectively.

3. The study involving chemical inhibition of NONO for telomerase recruitment analysis was performed in 293T cells (Fig. S5). To illustrate the broader application of this strategy, the chemicals should be applied to both 293T and HT1080 cells for a telomere length study. Additionally, it is crucial to determine if the chemicals affect Coilin and hTR levels.

We have now performed the telomerase recruitment experiments with the inhibitor in HT1080 cells (sFigure 7B-C). In addition, we have assayed Cajal body number, coilin integrated intensity per cell, and hTR integrated intensity per cell after treatment with (R)-SKBG-1 in 293T and HT1080 cells (sFigure 7D-F). Treatment with the inhibitor reduces telomerase recruitment in both cell lines but does not appear to drastically alter Cajal body number and hTR levels. These data are now included in the revised manuscript.

4. The authors contend that long-term NONO depletion impairs telomerase recruitment and causes telomere shortening (Line 302), based on studies in HT1080, HeLa, and HCT116 cells. However, the studies on HeLa and HCT116 cells are limited to TRF assays. To support this claim, the effects of NONO depletion on telomerase recruitment and Cajal body retention in HeLa and HCT116 cells should be examined.

To address this comment, we have performed hTR recruitment assays in long-term shRNA depleted HeLa and HCT116 cells showing impaired telomerase recruitment and Cajal body retention. These data are included in sFigure 8E of the revised manuscript.

5. In ChIP experiments, non-specific antibody binding can lead to false positive results. To demonstrate antibody specificity, a sample with depletion of the target protein should be included in the ChIP experiments. Unfortunately, this control was excluded from the ChIP experiments shown in Fig. 2E. Be aware that using IgG as a control is not a suitable replacement.

The specificity of the DBHS antibodies used for ChIP has been validated using knockdown experiments (sFigure 5F).

Minor Issues:

1. My experience is that ectopically expressing TERT in U2OS cells is sufficient to restore their telomerase activity, suggesting the presence of hTR in U2OS cells. Perhaps the U2OS strain used in this study could be indeed negative for both hTR and hTERT. The authors should provide real-time RT-PCR data to support this.

We have performed qTRAP analysis in U-2 OS cells overexpressing hTERT and hTERT/hTR+. Telomerase activity was only observed in U-2 OS cells overexpressing hTERT/hTR+, suggesting that the strain we used is negative for hTR. These data are included here for the reviewers' consideration.

2. The sentences starting from Line 208 and Line 210 seem to contradict each other.

Thank you. We have amended this.

3. Due to the lack of evidence for a direct physical interaction between DBHS proteins and hTR, the model should be modified.

After addressing the first concern raised by Reviewer 3, we now have direct evidence that all three DBHS proteins interact with hTR and assembled telomerase.

Reviewer #4 (Remarks to the Author):

In this manuscript, Sobinoff and colleagues report a novel role for the RNA-binding proteins NONO, SFPQ and PSPC1, which are part of the DBHS family, in telomerase trafficking and recruitment at telomeres. The authors provide evidence that members of the DBHS family are associated with the catalytically active telomerase and with telomeres. Furthermore, in the absence of DBHS proteins, telomerase RNA accumulates in Cajal bodies, and the recruitment of telomerase at telomeres is disrupted, leading to telomeres shortening. The authors propose that DBHS proteins are novel components of the telomerase trafficking machinery.

Overall, the authors have not convincingly showed that the effects observed on telomerase recruitment and telomeres length upon depletion of NONO, SFPQ or PSPC1 are direct and are not due to indirect effects, as these proteins play wide roles in RNA metabolism, including splicing and translation. For instance, the authors have not looked at the impact of NONO, SFPQ and PSPC1 depletion on the expression of the Shelterin proteins, especially

TPP1 and TIN2, which promote telomerase recruitment at telomeres. More important, several of the phenotypes observed upon depletion of DBHS proteins (accumulation of hTR in Cajal bodies, decreased recruitment of hTR at telomeres, telomeres shortening), have been observed in TPP1 or TIN2 mutants (Abreu et al., MCB, 2010; Zhong et al., Cell, 2012). The impact of DBHS protein depletion on the expression and alternative splicing of these proteins should be determined.

To directly address this concern, we examined TPP1 and TIN2 protein levels upon DBHS protein depletion and found no changes in expression (sFigure 1M). Furthermore, the ability of NONO and PSPC1 overexpression to rescue hTR retention in Cajal bodies caused by SFPQ depletion, but not telomerase recruitment to the telomere, suggests that the DBHS proteins are interacting with telomerase away from the telomere (Figure 2B-C). If DBHS protein depletion was disrupting TPP1 and TIN2 function, the separation of decreased recruitment of hTR at telomeres and accumulation of hTR in Cajal bodies would not be expected as disruption of TPP1 and TIN2 causes both phenotypes.

More important, there are several contradictory results and inconsistencies presented in this manuscript. The impact of NONO, SFPQ and PSPC1 depletion on telomerase activity, recruitment or telomeres length not only varies among cell lines but also depends on the type of depletion used (siRNAs versus CRISPR KO), and varies among CRISPR clones (see comments below). This raises concerns regarding the specific effect of DBHS proteins function in telomerase activity at telomeres.

Transient depletion of all three DBHS proteins consistently affects telomerase recruitment in all cell lines tested (Figure 1H-I; sFigure 1G-H). Long term depletion of PSPC1 consistently affects telomerase recruitment and telomere length in all cell lines tested (shRNA and CRISPR) (Figure 1H-I; sFigure 1G-H; Figure 4B-C; sFigure 8A-B). Depletion of NONO consistently affects telomerase recruitment and telomere length in all cell lines tested (siRNA, CRISPR, shRNA) (Figure 1H-I; sFigure 1G-H; Figure 4A; 4C; sFigure 8D-E) excluding cells from a 293 background, where it consistently causes telomere lengthening via an unknown mechanism. Long term depletion of SFPQ is not physiologically possible (lethality).

Finally, the claim that DBHS proteins are involved in telomerase trafficking is not supported by the data. As mentioned above, disruption of the TPP1-hTERT interaction could explain several of the results. The authors do not provide evidence for a direct role for DBHS proteins in telomerase RNA trafficking.

Our data provide direct evidence that all three DBHS proteins interact with active telomerase via hTR, and that DBHS protein depletion reduces telomerase recruitment to the telomere and alters telomere length. Furthermore, we now show that this is not the result of altered TPP1 and TIN2 protein levels (sFigure 1M). Overall, we believe this provides strong evidence that the DBHS proteins are involved in telomerase recruitment to the telomere.

Major comments:

- Figure 1A: following an IP of NONO, the authors claim that all three proteins (NONO, SFPQ and PSPC1) interact with hTR. Since NONO interacts with the two other proteins independently of hTR, this experiment does not support their claim. The authors have not performed co-IPs with PSPC1 or SFPQ to detect their interaction with hTR.

We have now performed Co-IP experiments with all three DBHS proteins confirming their interaction with hTR and active telomerase (Figure 1A; 1C; sFigure 1E).

- Figure 2: The authors should check if the association of the DBHS proteins at telomeres is RNA-dependent or not. These proteins are known to bind TERRA and RNA:DNA hybrids at telomeres (Petti et al., Nature Communication, 2019). RNase-treatment prior the ChIP assay would help answer this question.

RNase treatment will disrupt or degrade hTR and the telomerase complex, making the interpretation of results difficult. Petti et al. showed that depletion of NONO or SFPQ causes increased RNA:DNA hybrids, a substrate for DBHS binding at the telomere. However, despite increased substrate, we show that depletion of NONO or SFPQ impairs the ability of other DBHS proteins to interact with the telomere, suggesting that the interaction is not completely dictated by TERRA.

- The results reported are confusing. While depletion of NONO, SFPQ or PSPC1 by siRNA decreases hTR/telomeres colocalization in 293T cells (Figure 1G), deletion of NONO with Cas9 in 293T cells does not affect hTR/telomeres colocalization (Figure S6B) and results in longer telomeres in these cells (Figure S6C and D).

The results are predominantly consistent across cell lines, with opposing findings only for NONO depletion in 293-derived cells. We pursued multiple lines of investigation to determine why telomeres lengthen in 293/293T cells following NONO depletion, but were unable to identify a tangible explanation. Specifically, telomere lengthening was not due to ALT activation, increased telomerase recruitment, or increased telomerase activity (sFigure 9). Nevertheless, we feel that these data are important to report both in this context, as well as for the broader understanding of telomere length regulation, which is a highly complex process that is frequently simplified in the literature. We have amended the text to clarify these points.

Figure S6E: The two CRISPR clones of 293T deleted of NONO (clone 5 and clone 11) have opposite effect on telomerase activity (in one clone it goes up and in the other clone it goes down). Both clones show the same level of recruitment of hTR at telomeres, similar to the control cells. However, both clones show telomere elongation over time! These results are contradictory.

Telomerase activity has been shown to vary considerably between clonal populations in cancer cell lines⁹. Therefore, changes in telomerase activity observed between parental and DBHS CRISPR clones may be due to clonal variation rather than the loss of DBHS expression. This has now been discussed in the revised manuscript.

- Telomerase activity using qTRAP should be performed on the CRISPR clones of NONO and PSPC1 to determine if these KOs affect telomerase activity.

qTRAP results for the CRISPR clones of NONO and PSPC1 are now included in sFigure 9A.

Figure S2B: hTR levels are normal in 293T cells depleted of NONO, SFPQ or PSPC1, but they are low in HT1080 cells with the same depletions.

This is correct for NONO and SFPQ, but does not reach significance for PSPC1 (sFigure 4B). As mentioned in the manuscript, the DBHS proteins NONO and SFPQ may impact hTR stability in certain cell lines.

Figure S2E: depletion of NONO and SFPQ using siRNAs increases telomerase activity in 293T cell, but decreases telomerase activity in HT1080 cells.

As mentioned in the manuscript, the changes in telomerase activity are directly proportional to changes in hTR levels.

- In 293T cells treated with siRNA against PSCP1, hTR levels and formation of Cajal bodies is very low. Why?

The levels of hTR are not significantly different in 293T cells treated with PSCP1 siRNA (sFigure 4B). However, coilin levels are significantly reduced in siPSCP1 treated cells, which impacts hTR foci and Cajal body formation (sFigure 1I).

- Figure S5: Have you measured hTR levels in cells treated with the SKBG-1 drug? Have you validated the activity of the drug on the formation of NONO aggregates? How do you explain that the drug does not increase the accumulation of hTR in Cajal bodies, as the siNONO does?

We have now repeated the telomerase recruitment experiments with the inhibitor in HT1080 cells (sFigure 7B-C). In addition, we have also assayed Cajal body number, coilin integrated intensity per cell, and hTR integrated intensity per cell after treatment with (R)-SKBG-1 in 293T and HT1080 cells (sFigure 7D-F).

The drug was originally described in Kathman et al., where they showed it to cause the formation of dysfunctional NONO nuclear aggregates⁸.

Depletion of a protein (siNONO) is not always expected to produce the same results as chemical inhibition ((R)-SKBG-1). This is because the protein is still present with chemical inhibition, and can therefore exert an effect on the cell. (R)-SKBG-1 is an electrophilic small molecule that covalently binds NONO C145. This amino acid is unique to NONO and is located within the hinge region between the RMM1 and RMM2 DBHS domains⁸. We found that n.ΔRRM2 293T and n.ΔRRM1 HT1080 expressing cells with depleted endogenous NONO also caused reduced telomerase recruitment without affecting hTR accumulation in Cajal bodies (Figure 3B; sFigure 6D). This would suggest these domains are not necessarily required for NONO mediated trafficking out of Cajal bodies but are required for effective telomerase recruitment. (R)-SKBG-1 targets near this domain and phenocopies these results. This has now been discussed in the revised manuscript.

Minor comments:

- Figure 1C: why NONO needs to be overexpressed in cells for the qTRAP assay?

IP-TRAP requires the purification of telomerase associated proteins under non-denaturing conditions to preserve telomerase enzymatic activity. We utilised an anti-Myc tag antibody to IP exogenously expressed c-Myc tagged DBHS proteins, which is why we used overexpressed NONO. This allowed us to competitively elute DBHS complexes with c-Myc peptide under non-denaturing conditions to preserve associated telomerase complex activity.

- Legend of Figure 1: The legend mention that hTR was detected by Northern blot. However, dot blots are presented in this figure. This should be corrected.

We have changed “Northern blot” to “Northern dot blot” throughout the manuscript.

- Figure 2: ChIP should be reported as % of input. The current labeling of the Y axis “Telomeric DNA normalized to input (AU)” is confusing.

This has been corrected.

References

- (1) Venteicher, A. S.; Abreu, E. B.; Meng, Z.; McCann, K. E.; Terns, R. M.; Veenstra, T. D.; Terns, M. P.; Artandi, S. E. A human telomerase holoenzyme protein required for Cajal body localization and telomere synthesis. *Science* **2009**, *323*, 644-648.
- (2) Fuentes-Iglesias, A.; Garcia-Outeiral, V.; Pardavila, J. A.; Wang, J.; Fidalgo, M.; Guallar, D. An Optimized Immunoprecipitation Protocol for Assessing Protein-RNA Interactions. *STAR Protoc* **2020**, *1* (2). DOI: 10.1016/j.xpro.2020.100093.
- (3) Xi, L.; Cech, T. R. Inventory of telomerase components in human cells reveals multiple subpopulations of hTR and hTERT. *Nucleic Acids Res* **2014**, *42*, 8565-8577.
- (4) Chen, L.; Roake, C. M.; Freund, A.; Batista, P. J.; Tian, S.; Yin, Y. A.; Gajera, C. R.; Lin, S.; Lee, B.; Pech, M. F.; et al. An Activity Switch in Human Telomerase Based on RNA Conformation and Shaped by TCAB1. *Cell* **2018**, *174* (1), 218-230.e213. DOI: 10.1016/j.cell.2018.04.039.
- (5) Cohen, S. B.; Graham, M. E.; Lovrecz, G. O.; Bache, N.; Robinson, P. J.; Reddel, R. R. Protein composition of catalytically active human telomerase from immortal cells. *Science* **2007**, *315*, 1850-1853.
- (6) Petti, E.; Buemi, V.; Zappone, A.; Schillaci, O.; Broccia, P. V.; Dinami, R.; Matteoni, S.; Benetti, R.; Schoeftner, S. SFPQ and NONO suppress RNA:DNA-hybrid-related telomere instability. *Nat Commun* **2019**, *10* (1), 1001. DOI: 10.1038/s41467-019-08863-1.
- (7) Zhong, F. L.; Batista, L. F.; Freund, A.; Pech, M. F.; Venteicher, A. S.; Artandi, S. E. TPP1 OB-fold domain controls telomere maintenance by recruiting telomerase to chromosome ends. *Cell* **2012**, *150*, 481-494.
- (8) Kathman, S. G.; Koo, S. J.; Lindsey, G. L.; Her, H. L.; Blue, S. M.; Li, H.; Jaensch, S.; Remsberg, J. R.; Ahn, K.; Yeo, G. W.; et al. Remodeling oncogenic transcriptomes by small molecules targeting NONO. *Nat Chem Biol* **2023**, *19* (7), 825-836. DOI: 10.1038/s41589-023-01270-0.
- (9) Bryan, T. M.; Englezou, A.; Dunham, M. A.; Reddel, R. R. Telomere length dynamics in telomerase-positive immortal human cell populations. *Exp. Cell Res* **1998**, *239*, 370-378.

RESPONSE TO REVIEWERS

Referee #1:

Reviewer #1: The authors have made a significant effort to improve the original version of the manuscript. In particular, the new EMSA experiments are highly informative (note: the legend of Figure 1 lacks information on EMSAs). The authors have responded clearly to the majority of comments. The new title of the manuscript is more appropriate.

The information on the EMSAs has now been included in the figure legend.

I support publication, although the contradictory results on telomere length in 293 and 293T cells after long-term NONO loss of function experiments do not support the proposed model.

The proposed model (Figure 5) has been changed to focus more on PSPC1, for which depletion caused consistent telomere shortening in all cell lines tested (Figure 4B; sFigure 8A).

I appreciate the ALT validation in these long-term models; however, the manuscript does not provide a clear explanation for these contradictory results. Therefore, I suggest that the statement in line 445 of the discussion: "...these anomalous results highlight the complexity of telomere maintenance" should be improved to provide a better explanation or hypothesis.

This sentence has been changed to "These anomalous results highlight the complexity of telomere maintenance, as telomere lengthening was observed without altering traditional markers of telomere maintenance".

A comment on the abstract: the statement "...culminating in progressive telomere shortening in several cell lines" is misleading. It implies that telomeres shorten in the several cell lines tested; the statement does not clarify that in some cell lines the telomere length phenotype is different. This information should be included in the abstract, giving credit to the complex interplay of the DBHS proteins.

This statement has been changed to "...with NONO and PSPC1 depletion culminating in progressive telomere shortening in several cell lines, with the exception of long-term NONO depletion in 293 and 293T."

Referee #2:

Reviewer #2: I co-reviewed this manuscript with one of the reviewers who provided the listed reports. This is part of the Nature Communications initiative to facilitate training in peer review and to provide appropriate recognition for Early Career Researchers who co-review manuscripts.

Referee #3:

Reviewer #3: The revised manuscript provides substantial evidence supporting the interaction between DBHS proteins and telomerase. The study presents novel findings demonstrating that DBHS proteins directly regulate intracellular telomerase function, influencing telomere length homeostasis.

However, the varying effects of DBHS protein loss-of-function on telomere lengthening across different cell types suggest potential cell-type-specific roles for individual DBHS proteins, and thus the underlying mechanisms require further investigation.

We agree that further investigation is required to tease apart the cell line specific nature of long term NONO depletion on telomere length between cell lines. This will be addressed in future studies.

Referee #4:

Reviewer #4: In this revised manuscript, the authors provide new data regarding the role of DBHS proteins in telomeres biology. Overall, they have properly answered several of the comments and questions raised by the reviewers. This reviewer agrees with the authors saying, "What is clear, is that disrupting members of the DBHS protein family consistently impacts telomere length, and that PSPC1 is essential for telomere length maintenance by telomerase". Moreover, the authors now provide convincing evidence for DBHS proteins interaction with telomerase.

However, there is still no clear mechanism, and I am not convinced that the telomere phenotypes are due to a role of DBHS proteins in recruitment of telomerase to the telomeres, as the title of the manuscript suggests.

The authors put a lot of emphasis on the role of DBHS proteins in telomerase recruitment, but their data also point toward a role for these proteins in hTR expression and telomerase activity, especially in HT1080 cells (sFigure 4).

In figure 4, the authors have not measured hTR levels and telomerase activity in the NONO and PSPC1 KO HT1080 cells. Telomeres shortening in these KO cells may well be explained by reduced hTR levels. This may also explain the reduction in hTR/telomeres colocalization event per cell, as lower level of hTR would decrease the number of hTR/telomeres colocalization events.

We have now included telomerase activity (Supplementary Figure 9A) and hTR levels (Supplementary Figure 9B) in KO HT1080 cells. PSPC1 depletion led to an increase in telomerase activity, indicating that the reduction in telomere length in these cells is primarily due to decreased telomerase recruitment. In NONO-depleted cells, telomerase activity decreased—consistent with results from siRNA knockdown. However, while this reduction may contribute to the observed decrease in telomerase recruitment, it does not fully explain the reduction in recruitment.

Additionally, hTR levels in NONO and PSPC1 knockout cells do not positively correlate with the reduced telomerase recruitment, suggesting that telomere shortening is not solely due to lower hTR levels (Supplementary Figure 9B; Figure 4C; Supplementary Figure 8B).

hTR levels have not been quantified in HeLa and HCT116 cells depleted of NONO using shRNAs, so it is not clear that changes in hTR expression is a cell line specific effect, as claimed by the authors.

We have now included hTR integrated intensity per cell in HeLa and HCT116 cells where NONO was depleted using shRNA (Supplementary Figure 8F). While a slight decrease in hTR levels was observed in the shNONO-depleted cells, this reduction is insufficient to explain the observed decrease in telomerase recruitment (Supplementary Figure 8E).

In general, quantification of hTR RNA level should be done more thoroughly in the manuscript. hTR RNA was quantified using dot blots, which cannot distinguish between hTR size variants (mature versus unprocessed). Furthermore, no loading control was included in the input of the dot blots (like Actin mRNA or U1 snRNA) in sFigure 4A-B, so it is not clear

how accurately hTR level is measured with these blots. Northern blot or RT-qPCR would be more appropriate.

To address the accuracy in hTR level detection, we initially measured total hTR levels in siDBHS 293T cells using three different methods: dot blot (data shown in Supplementary Figure 4A-B; Figure A, below), hTR integrated intensity per cell (Figure B, below), and Northern blot (mature hTR; Figure C, below). The results from all three techniques were consistent with each other.

For this reason, we progressed with the dot blot technique (Supplementary Figure 4A-B). This is because this method enables direct comparison between total hTR (input) and the hTR bound to hTERT after immunoprecipitation (assembled telomerase). Using hTR qPCR on hTERT IP samples would not allow for the simultaneous assessment of protein levels and telomerase activity. Elution in Buffer A enables multiple analyses from the same sample: western blot, hTR dot blot, and qTRAP assays.

RESPONSE TO REVIEWERS

Referee #4:

Reviewer #4: By including their new data, the authors have properly answered my comments in this revised version of their manuscript. They also clarified some comments in the text. Although there are still questions regarding the cell line specific effect of the DBHS proteins on telomere homeostasis, I support the publication of this manuscript in this journal.